# Myogenin promotes myocyte fusion to balance fibre number and size

Massimo Ganassi [1], Sara Badodi [2], Huascar Pedro Ortuste Quiroga [1,3], Peter S. Zammit [1], Yaniv Hinits [1] & Simon M. Hughes [1]

Each skeletal muscle acquires its unique size before birth, when terminally differentiating myocytes fuse to form a defined number of multinucleated myofibres. Although mice in which the transcription factor Myogenin is mutated lack most myogenesis and die perinatally, a specific cell biological role for Myogenin has remained elusive. Here we report that loss of function of zebrafish *myog* prevents formation of almost all multinucleated muscle fibres. A second, Myogenin-independent, fusion pathway in the deep myotome requires Hedgehog signalling. Lack of Myogenin does not prevent terminal differentiation; the smaller myotome has a normal number of myocytes forming more mononuclear, thin, albeit functional, fast muscle fibres. Mechanistically, Myogenin binds to the *myomaker* promoter and is required for expression of *myomaker* and other genes essential for myocyte fusion. Adult *myog* mutants display reduced muscle mass, decreased fibre size and nucleation. Adult-derived *myog* mutant myocytes show persistent defective fusion ex vivo. Myogenin is therefore essential for muscle homeostasis, regulating myocyte fusion to determine both muscle fibre number and size.

[1] Randall Centre for Cell and Molecular Biophysics, King's College London, London SE1 1UL, UK. [2] Blizard Institute, Barts and The London School of Medicine and Dentistry, Queen Mary University of London, 4 Newark Street, London E1 2AT, UK. [3] Present address: School of Health Sciences, Toyohashi Sozo University, Matsushita-20-1 Ushikawacho, Toyohashi, Aichi Prefecture 440-0016, Japan. Correspondence and requests for materials should be addressed to S.M.H. (email: s.hughes@kcl.ac.uk)

Regulation of tissue size requires balancing cell number and cell size. In skeletal muscle, tissue size depends on generating the correct number of multinucleated muscle fibres with an appropriate number of nuclei in each; how these processes are controlled in vertebrates is mysterious. Formation of syncytial muscle fibres is a three-step process: commitment as a myoblast, terminal differentiation into a myocyte, defined here as irreversible cell cycle exit and expression of muscle-specific actin, myosin and other genes, and finally myocyte fusion and growth to form a mature multinucleate myofibre. In specialised circumstances, such as the mononucleate slow myofibres of larval zebrafish, myocytes mature into functional innervated and contractile muscle fibres without fusion. The Myogenic Regulatory Factor (MRF) family of transcription factors (Myod, Myf5, Mrf4 and Myog) are key players in orchestrating each of these steps in skeletal myogenesis[1,2]. All MRF genes encode a basic domain and Helix–Loop–Helix (bHLH) motif, which account for protein–DNA binding and hetero-dimerization with ubiquitous E-proteins, respectively, by which they activate expression of many E-box-containing muscle-specific genes[3,4]. Whereas three MRFs drive myoblast formation during early development, Myog acts later to regulate myoblast terminal differentiation, myofibre maturation and size[1,2]. Genetic studies in mice revealed that among MRFs only Myog is essential for viability; null Myog mutation leads to perinatal death, due to severely defective muscle differentiation, although residual differentiated muscle fibres are present[5–8]. Absence of Myog does not prevent slow and fast fibre type diversification[7,8]. In vitro studies on primary myoblasts or embryonic stem cells from Myog mutant mice reveal terminal differentiation comparable to wild-type (wt) albeit yielding smaller syncytial myotubes[6–10], suggesting that extracellular factors determine the need for Myog function. Congruently, $Myog^{-/-}$ myoblasts efficiently contribute to multinucleated fibres in genetic mosaic experiments[11]. Furthermore, depletion of Myog after birth reduces myofibre size and affects overall body homeostasis, although without perturbing muscle histology[12–14]. However, given that other MRFs can and do bind the same DNA motifs as Myog[15], the precise role(s) of Myog remain ill-defined.

Knockdown of zebrafish myog has minor effects on initial events in myogenesis[16–19], whereas combined knockdown of Myog and Myod strongly reduces myogenesis of fast myofibres[17]. These findings were confirmed using a zebrafish mutant ($myog^{fh265}$) bearing a stop mutation downstream of the bHLH domain, which also shows delayed muscle regeneration[18,20]. However, as a similar truncation of mouse Myog has residual activity, we previously suggested that $myog^{fh265}$ is hypomorphic[18,21]. Hence, despite strong evidence for roles for Myog in later myogenesis not compensated for by other MRFs[7,8], a specific evolutionarily-conserved function in vivo is unclear.

Here we create null alleles in zebrafish myog and reveal a specific function for Myog in myocyte fusion during skeletal muscle development. We find that Myog is dispensable for myoblast terminal differentiation, expression of many muscle-specific markers, myofibre elongation across the somite, sarcomere assembly, innervation and generation of functional contractile muscle. However, lack of Myog prevents most myocyte fusion and leads to supernumerary mononucleated muscle fibres. Myog is required for the expression of membrane proteins involved in cell fusion, such as Myomaker[22–25]. Despite gross myocyte fusion defects, zebrafish myog mutants survive to adulthood with more but thinner muscle fibres and reduced overall body size. Adult muscle precursor cells lacking Myog show a persistent fusion defect ex vivo. Interestingly, residual fusion in myog mutants occurs primarily in the deep myotome and is dependent upon Hedgehog signalling, indicating the existence of two pathways to myocyte fusion.

## Results

**Generation of myogenin mutant alleles.** To create a null myog mutant, we targeted genome editing far upstream of bHLH region and obtained two nonsense alleles (Fig. 1a). $Myog^{kg128}$ has an insertion of 1 bp (A), whereas $myog^{kg125}$ has a deletion of 3 bp (TCA). Both mutations create a stop codon in an identical position (Y37*), producing a truncated protein lacking both basic and HLH domains. In situ mRNA hybridisation (ISH) for myog on $myog^{kg128/+}$ and $myog^{kg125/+}$ incross lays at 18 h post fertilization (hpf) showed reduced signal in mutant embryos compared to siblings (sibs), presumably by nonsense-mediated decay (NMD) (Fig. 1b). mRNA downregulation was confirmed by qPCR at 20 hpf (Fig. 1c). Congruently, mutant embryos lacked Myog immunoreactivity, whereas F-actin accumulation and overall number of nuclei per myotome was indistinguishable from wild-type (wt) (Fig. 1d). Heterozygote and wt siblings showed similar levels of Myog (Supplementary Fig. 1a). No compensatory upregulation of other MRFs was noted at 20 hpf. Indeed, lack of Myog significantly reduced expression levels of myf5 (40%) and mrf4 (54%), whereas myod remained unchanged (Fig. 2a). These results demonstrate that homozygous mutant alleles $myog^{kg125}$ or $myog^{kg128}$ block Myog mRNA and protein accumulation.

**Myoblasts differentiate and muscle functions without Myogenin.** To examine muscle differentiation, mutants and sibs were compared for mylpfa and smyhc1 expression that distinguish fast and slow muscle[26–28]. ISH analysis revealed no difference at 15 somite stage (ss) (Supplementary Fig. 1b). In this and subsequent experiments, no differences were observed between wild type and heterozygous sibs, consistent with their similar Myog level (Supplementary Fig. 1a). At 22 ss, strong mylpfa mRNA in fast muscle in anterior somites and smyhc1 mRNA in slow muscle extending more posteriorly were also unaltered in myog mutants (Supplementary Fig. 1c). At this stage, slow myofibres have migrated to the lateral surface of the myotome and remain mononucleated, whereas the more abundant multinucleated fast muscle fibres are located deeper in the myotome[29–31]. Moreover, no obvious difference was observed, either in motility or in fast and slow myosin heavy chain (MyHC) immunoreactivity at 20 hpf, 1 day post-fertilisation (dpf) or 2 dpf, when embryos have hatched and make short bursts of controlled swimming (Fig. 2b,c and Supplementary Fig. 1c-e). Slow myofibre number and thickness were not affected in mutants (Fig. 2b and Supplementary Fig. 1e). Thus, without Myogenin, specification and early development of slow and fast muscles appears normal.

α-Actinin, Titin, F-actin and Acetylcholine Receptor staining also showed that fibre formation, sarcomere assembly and innervation had occurred properly in mutants (Fig. 2d–h). At 2 dpf, mutant myofibres were correctly positioned and elongated across the length of the somite. However, fast muscle appeared mildly disorganised and slightly reduced in extent (Fig. 2f). Nevertheless, motor function at 5 dpf assayed by time spent swimming, total travelled distance and average speed did not differ between myog mutants and sibs. Irrespective of genotype, some larvae were consistently more active than others throughout the 30 min measurement period (Fig. 2i). To test fibre integrity and anchorage, fish were swum in a viscous methyl-cellulose (MC) solution, which led to a general decrease of swimming performance. Despite this challenging environment, mutants did not perform significantly worse than their siblings (Fig. 2i). When sib and mutant larvae were grown in MC from 5 to 8 dpf, a procedure known to damage defective muscle[32], myog mutants retained good muscle morphology (Supplementary Fig. 1f). Thus, Myogenin is dispensable for the initial phases of myogenesis and generation of strong functional muscle in zebrafish.

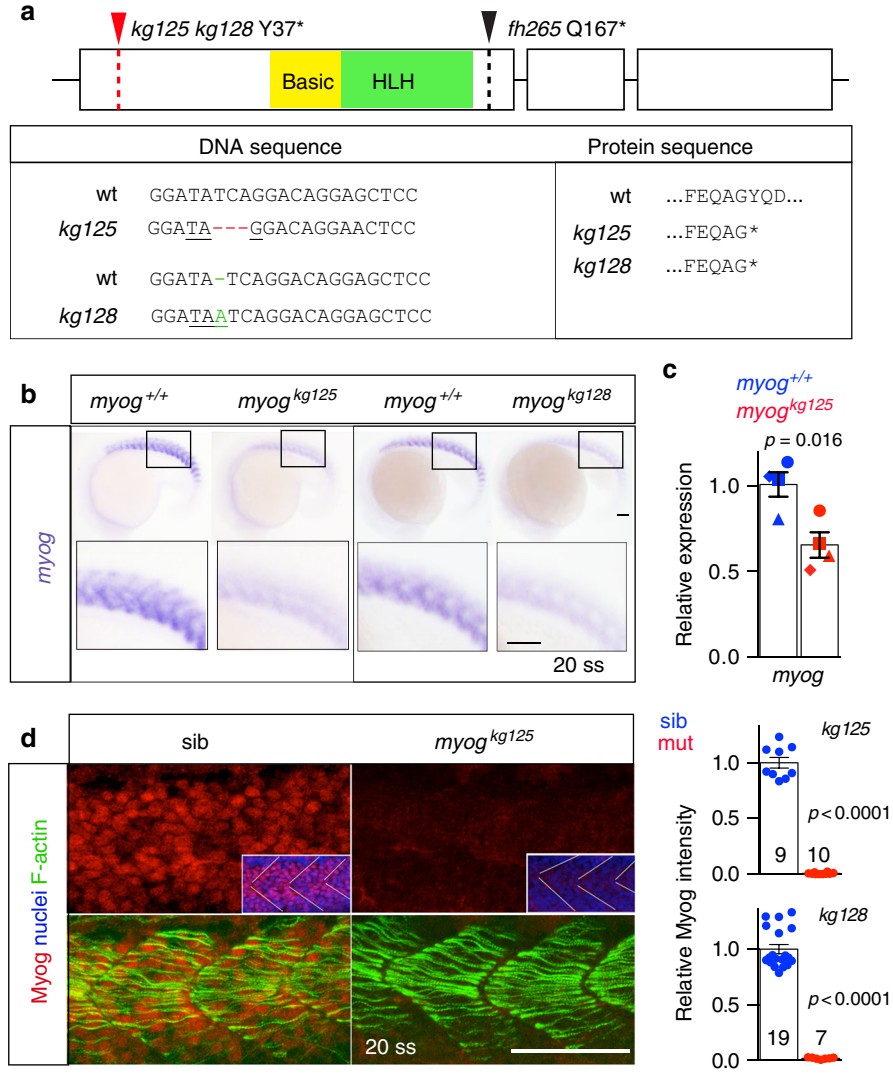

**Fig. 1** Genome editing generates zebrafish *myog* null alleles. **a** Schematic of *myog* gene exons (boxes) and protein showing the position of *kg125*, *kg128* and *fh265* mutant alleles. The tyrosine to stop mutations (Y37*) produce a truncated protein of 36 amino acids (aa) devoid of both basic (yellow) and helix–loop–helix (HLH, green) domains. The *fh265* hypomorphic allele (Q167*) truncates downstream of bHLH. Beneath, DNA and protein alignment of wild-type (wt), 3 bp *kg125* deletion (red) and 1 bp *kg128* insertion (green) alleles with novel stop codons underlined causing an identical truncation. **b** In situ mRNA hybridisation (ISH) for *myogenin* on *myog*<sup>kg128/+</sup> and *myog*<sup>kg125/+</sup> incross lays reveal NMD of mutant *myog* mRNA at 20 somite stage (20 ss). Representative images $n = 14 + 5$ mutants, $n = 39 + 22$ sibs, respectively. Insets show magnification of boxed areas. **c** qPCR analysis on wt sibs and *myog*<sup>kg125</sup> embryos at 22 ss confirms NMD. Mean fold change ± SEM from four independent experiments on genotyped embryos from four separate lays analysed on separate days, paired *t* test statistic. Symbol shapes denote matched wt and mutant samples from each experiment. **d** Immunoreactivity of Myog is lost in *myog*<sup>kg125</sup> and *myog*<sup>kg128</sup> mutants at 20 ss, whereas F-actin is unaffected. Insets show nuclear staining in *myog*<sup>kg125</sup> and sib using Hoechst counterstain. Relative myotomal Myog immunofluorescence was assessed by nuclear intensity measurement. All images are lateral views anterior to left, dorsal to top. Representative images $n = 10 + 7$ mutants, $n = 9 + 19$ sibs, respectively. Bars = 50 µm

**Muscle size reduction and myofibre number increase in *myog* mutant**. Although functional muscle was formed, our data suggested a reduction in myotome size in mutants (Fig. 2). Measurement of the dorso-ventral extent of ISH staining of fast *mylpfa* and slow *smyhc1* revealed a reduction of mutant myotome size, both at 1 and 2 dpf (Fig. 3a, b and Supplementary Fig. 2a). Nevertheless, the body length of mutants and sibs was comparable, suggesting that muscle reduction was not due to reduced overall body size or delayed development (Fig. 3c and Supplementary Fig. 2b).

To analyse the defect in cellular detail, *myog*<sup>kg128</sup> and *myog*<sup>kg125</sup> were bred onto *Tg(Ola.Actb:Hsa.HRAS-EGFP)*<sup>vu119</sup>, in which EGFP targets plasma membranes of all cells[33] (*β-actin: EGFP* hereafter). Confocal sections of *β-actin:EGFP;myog*<sup>kg128/+</sup>

incross larvae confirmed that reduction in myotome volume was present at 2 dpf in mutants and persisted until at least 5 dpf (Fig. 3d, f). Although myotome cross-sectional area was consistently reduced in mutants, myotome length was unaffected (Fig. 3g; Supplementary Fig. 2b). A reduction of fast fibre cross-sectional area in mutant fish was observed (Fig. 3e). Indeed, quantitative analysis at 2 dpf revealed a 50% increase of fast fibre number accompanied by a 45% reduction in mean fibre volume in mutant embryos (Fig. 3h, i). *myog*<sup>fh265</sup> mutants did not have reduced size or altered cellularity and we did not observe any other phenotype in un-manipulated embryos, larvae or adults, confirming that this allele is hypomorphic[18] (Supplementary Figs 2c–e; 5a). We conclude that Myog controls fast myofibre number and size.

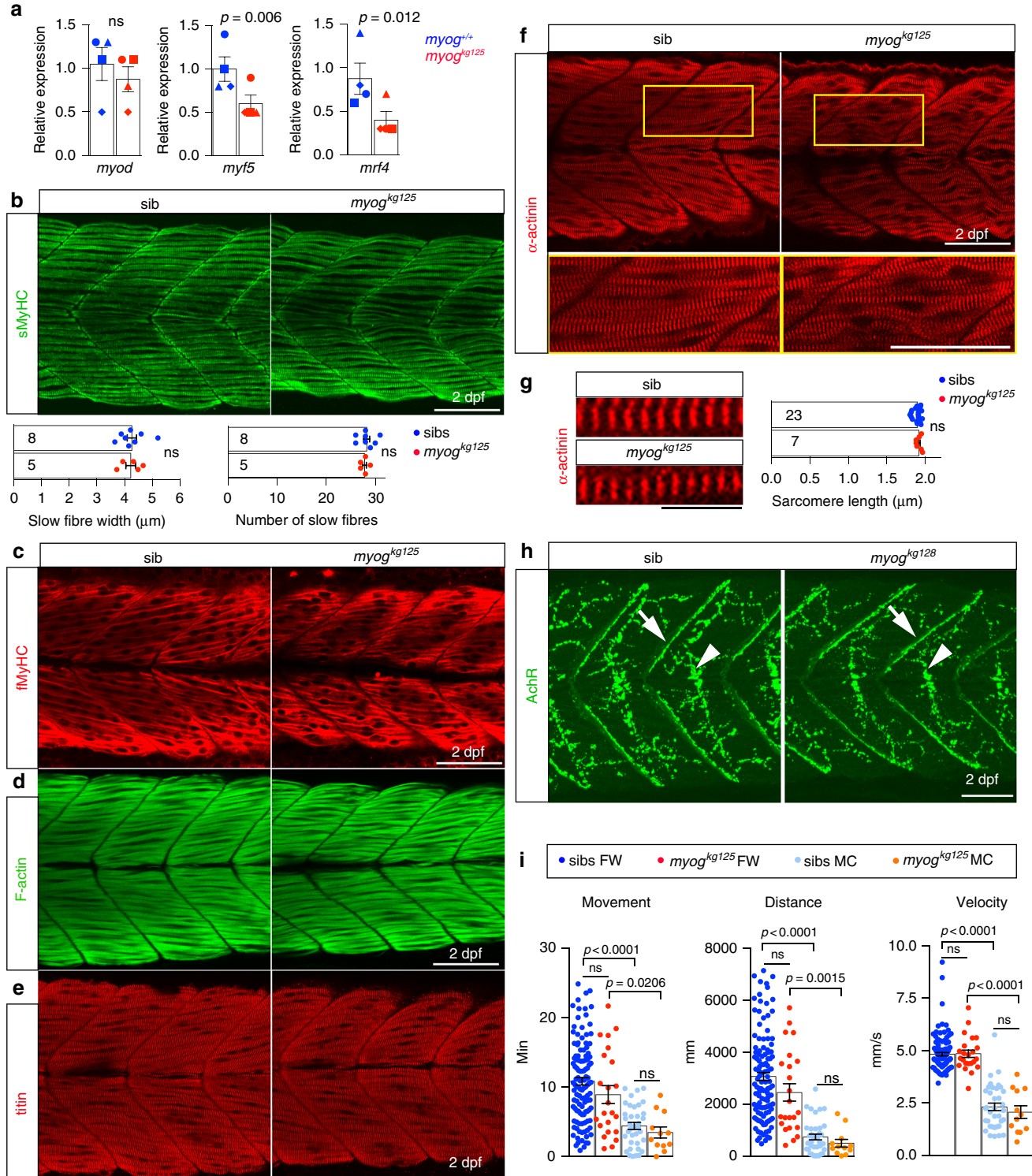

**Myogenin is required for normal myocyte fusion.** Two hypotheses could explain the overabundance of fast muscle fibres in *myog*[kg128] and *myog*[kg125] mutant embryos: increased frequency of terminal differentiation of muscle progenitor cells (MPCs) into myocytes or reduced fusion of differentiating myocytes into multinucleate fibres. Most, if not all, fast myofibre nuclei derive from MPCs expressing either Pax3 or Pax7[34–42]. We found no differences between *myog* mutants and sibs in position or number of Pax3/7 positive cells at 1–2 dpf (Fig. 4a, b). Nor was there a

change in the number of nuclei in the myotome (Fig. 4h). Thus, no evidence supported the possibility that myocyte formation from MPCs was increased.

Next, we tested the ability of Myog to promote myocyte fusion. We injected DNA encoding CAAX-mCherry at the 1-cell stage into *myog*[kg128/+] incross embryos to label mosaically the plasma membrane of single fibres. At 2 dpf the majority of labelled fibres in mutant embryos were mononucleated, whereas those in sibs were multinucleate (Fig. 4c). To quantify this defect, we injected

**Fig. 2** Functional muscle differentiation in Myogenin mutants. **a** qPCR analysis on wt sibs and *myog*[kg125] embryos at 22 ss showing reduction of *myf5*, *mrf4* but not *myod* RNAs. Mean fold change ± SEM from four independent experiments on genotyped embryos from four separate lays analysed on separate days, paired *t* test statistic. Symbol shapes denote matched wt and *myog* mutant samples from each experiment. **b**, **c** Immunodetection of slow and fast myosins (sMyHC and fMyHC) in 2 dpf larvae from a *myog*[kg125/+] incross showing that myofibre differentiation occurs in mutant. Dots in graphs show slow myofibre width (average of six myofibres/larva) and number in somite 17 of sib and mutant individuals. Representative images *n* = 5 mutants, *n* = 8 sibs. **d**–**f** Phalloidin staining for F-actin or immunolabelling for titin and α-actinin reveals that mutant myofibres display regular sarcomere spacing and are properly assembled to sustain contraction. Representative images *n* = 5 mutants, *n* = 21 sibs (phalloidin); *n* = 7 mutants, *n* = 18 sibs (titin); *n* = 7 mutants, *n* = 23 sibs (α-actinin). In **f**, boxes are shown magnified below. **g** Sarcomere length from **f** (average of 6 myofibres/larva) in sibs and mutants. Numbers of larvae analysed are shown on columns (**b**, **g**). **h** Larvae from a *myog*[kg128/+] incross stained with α-bungarotoxin-Alexa-488 show that mutant embryos accumulate AChR at both neuromuscular junction (arrows) and muscle-muscle junction (arrowheads) comparable to sibs. Representative images *n* = 4 mutants, *n* = 11 sibs. **i** Motor function of 5 dpf larval zebrafish in fish-water (FW; *n* = 24 mutants, *n* = 115 sibs) or 0.6% Methyl-Cellulose (MC; *n* = 12 mutants, *n* = 36 sibs) as time spent moving (minutes), distance travelled (mm) and average speed (mm/s). Overall muscle function is unaffected by lack of Myog in both FW and MC. Activity of both *myog* mutants and sibs is affected by MC. Each dot represents the behaviour of an individual larva. ns: not statistically significant in ANOVA. Bars = 50 μm (10 μm in **g**)

H2B-mCherry mRNA into *β-actin:EGFP;myog*[kg125] embryos to label nuclei and plasma membranes uniformly. Strikingly, over 92% of muscle fibres in *myog* mutants were mononucleate, compared to 33% in sibs (Fig. 4d, e; Supplementary Fig. 3a). The fraction of nuclei in multinucleate fibres dropped from 81% in sibs to 16% in mutants (Fig. 4f; Supplementary Fig. 3a). This phenotype persisted at least until 6 dpf (Supplementary Fig. 3b). Nuclei in mononucleated myofibres preferentially located near the centre of the myotome (Fig. 4d). Despite the twofold reduction in myonuclei per myofibre, the total number of nuclei within in each myotome was not altered in mutants, paralleling the increase in myofibre number (Figs. 3h; 4g, h).

To eliminate the possibility of CRISPR off-target effects, we re-expressed Myog mosaically in *myog* mutant larvae by injection of a plasmid containing the zebrafish *myog* promoter driving wt zebrafish *Myog-IRES-GFP* expression. *myog:MyogCDS-IRES-GFP* rescued fusion in fast fibres, whereas a control *myog:GFP*-only vector did not (Fig. 4i, j). Myog knockdown with a morpholino fully recapitulated the mutant phenotype with increased number of mononucleate fibres and decreased somite growth (Supplementary Fig. 3c–e). We conclude that Myogenin is essential for most myocyte fusion.

To examine the cell autonomy of the need for Myogenin we analysed *myog* mutant larvae with mosaic *Myog-IRES-GFP* re-expression further. *Myog-IRES-GFP* significantly rescued fusion; Myog-expressing fibres in *myog* mutants contained a range of nuclear numbers approaching the distribution in controls (compare Fig. 4e, i). Adjacent unlabelled fibres remained mononucleate (Fig. 4j). As cells expressing *Myog-IRES-GFP* occurred at a rate of about 2–3 per somite and were well-scattered, this finding indicates that only a single fusing partner needs to express Myog to permit fusion. Moreover, although mosaic Myog expression rescued mutants, it did not induce more fusion than observed in wt in either mutants or sibs (Fig. 4i, j; Supplementary Fig. 4a–c). Importantly, Myog expression (either in *myog* mutants or sibs) failed to elicit fusion of the normally-mononucleate slow fibres, even though the slow fibres were adjacent to fast fibres and their MPC precursors, indicating that the low level of *myog* mRNA in wt slow MPCs is not the only reason for their lack of fusion (Supplementary Fig. 4d,e). However, both mutant and sib slow fibres overexpressing *Myog-IRES-GFP* showed significantly reduced myofibrillar width (Supplementary Fig. 4e,f). Thus, Myog is required in at least one of two fusing fast myocytes to permit fusion.

**Expression of fusogenic genes reduced in *myog* mutants.** Fusion of myocytes is a key feature of skeletal myogenesis and requires several transmembrane proteins[43]. We hypothesised that Myog regulates these genes. *Myomaker* (*mymk*)[22–25,44] mRNA was

strongly reduced (72%) in *myog* null mutant embryos at 20 hpf, during initial myocyte fusion, and was also mildly affected in *myog*[fh265] hypomorphs (Fig. 5a, b; Supplementary Fig. 5a), paralleling the previously observed *myog* nonsense-mediated mRNA decay in this hypomorphic allele[18]. *Mymk* was also reduced in *myod*[fh261] mutant in proportion to *myog* mRNA reduction and loss of fast muscle[18,45] (Supplementary Fig. 5a,b). Reduction of *mymk* mRNA thus parallels lack of myocyte fusion. *Myomixer/ myomerger/minion*, a micropeptide recently described to enhance myoblast fusion[44,46,47], was also reduced (34%; Fig. 5b). Moreover, *jam3b* mRNA was significantly reduced (22%) in mutants, but *jam2a*[48] and *kirrel3l*[49] were unaffected (Fig. 5a, b).

The extent of reduction of *myomaker* expression in mutants argues for direct transcriptional regulation by Myogenin. To test whether Myog directly regulates *mymk* transcription in zebrafish, we scanned 3 kb of putative promoter region upstream of the *myomaker* 5'-UTR and found two E-box elements (E-box 1 and E-box 2, Fig. 5c). ChIP-qPCR assay on 20 hpf embryos revealed that endogenous Myog binds both E-box elements, with significant enrichment of Myogenin binding to E-box 1 compared to two different negative controls. The more proximal E-box 1 showed greater binding than E-box 2 (Fig. 5d). Combined, these data support a role for Myogenin in governing myocyte fusion through direct transcriptional upregulation of *mymk* and other fusogenic factors.

**Hedgehog drives residual fusion and *mymk* expression.** *Myog* mutants retain small numbers of multinucleate fibres in the medial somite (Fig. 4e; Supplementary Fig 3a,b). Residual *mymk* mRNA also persists in *myog* mutants (Fig. 5a, b), showing that other factors drive *mymk* expression in some cells. Residual *mymk* mRNA is preferentially enriched in the medial region of mutant myotome, adjacent to the notochord (Fig. 6a). Notochord-derived Hedgehog (Hh) signals promote differentiation of slow and a medial subset of fast muscle in zebrafish[19,30,35,37,50–54]. Treatment of *myog* mutant embryos with the Hh inhibitor cyclopamine (CyA) led to an additional 54% reduction of *mymk* mRNA, leaving < 20% of the original *mymk* expression compared to vehicle-treated wt siblings (Fig. 6b; Supplementary Fig. 5c,d). CyA-treated sibs also showed a 22% *mymk* reduction (Supplementary Fig. 5c–e) compared to controls. Congruently, when a *β-actin:EGFP;myog*[kg125/+] incross was treated with CyA, residual fusion in *myog* mutants at 2 dpf was largely lost (Fig. 6c). Blockade of Hh signalling had no detectable effect on fusion in sibs, although reducing both sib and mutant fast muscle growth, as previously reported[55] (Fig. 6c). These observations show that in *myog* mutants Hh signalling sustains residual *mymk* expression and myocyte fusion in the deep/medial myotome close to the notochord.

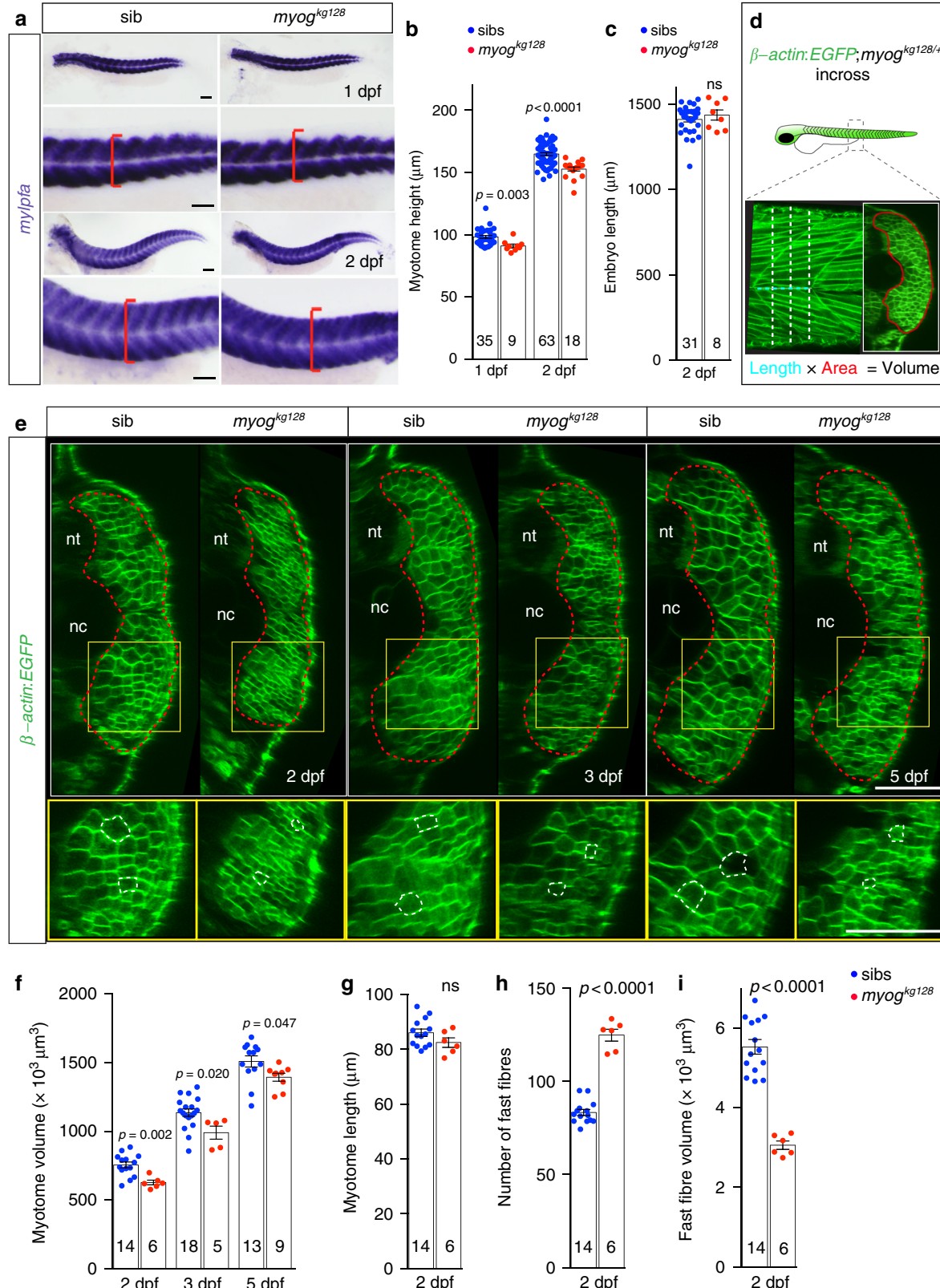

**Adult *myog* mutants have small muscle with reduced fibre size**. Both putative null *myog* alleles are homozygous viable. By 4 months (120 dpf), compared to their co-reared sibs, adult mutants showed a reduction in standard weight, a measure that compensates for length changes (Fig. 7a, b; Supplementary Fig. 6a, b). In contrast, *myog^fh265^* mutants were similar to sibs,

consistent with the lack of larval phenotype (Fig. 7b; Supplementary Fig. 6a). The new mutants showed a 35-40% reduction in weight and lower 'body mass index'. These data show that muscle bulk reduction is independent of, and may cause, the observed length reduction (Supplementary Fig. 6a). Muscle reduction persisted also in 15 month old mutants (Supplementary Fig. 6c).

**Fig. 3** Myogenin is required for normal larval muscle growth. **a, b** ISH for *mylpfa* mRNA showing similar level of expression of fast myosin but reduced extent of somitic muscle in mutants (**a**, red brackets) at 1 and 2 dpf. Myotome height (**b**) is significantly reduced in *myog*[kg128] mutants at 1 and 2 dpf. Representative images $n = 9 + 18$ mutants, $n = 35 + 63$ sibs. **c** Larval length is unaffected in mutant at 2 dpf, showing that muscle reduction is not due to overall reduced size of mutant larvae. **d** Schematic of myotome volume measurement. *β-actin:EGFP;myog*[kg128/+] fish were incrossed and progeny imaged by confocal microscopy. Lateral and three equi-spaced transverse images of somite 17 were collected from each larva at each stage (dashed lines). Transverse area (red outline) multiplied by somite length (cyan dashed line) yielded myotome volume for each fish at each stage. **e** Optical transverse-sections of *β-actin:EGFP;myog*[kg128] mutants show reduced myotome area at 2, 3 and 5 dpf compared to sibs (dashed red lines). Boxes are shown magnified beneath highlighting smaller fibre cross-sections in mutants (dashed white lines). Representative images $n = 6$ mutants, $n = 14$ sibs (2 dpf); $n = 5$ mutants, $n = 18$ sibs (3 dpf); $n = 9$ mutants, $n = 13$ sibs (5 dpf). **f** Myotome volume reduction in *myog*[kg128] mutants at 2, 3 and 5 dpf. **g–i**. Myotome length (**g**), number of fast fibres per cross section (**h**) and fibre volume (**i**) at 2 dpf compared by *t* tests. Bars = 50 μm. nt: neural tube, nc: notochord

Adult muscle phenotype examined in transverse cryosections of 120 dpf *myog*[kg128] incross fish revealed a decreased myotome area, reduced fibre cross-sectional area and increase in fibre number per unit area of muscle in *myog*[kg128] mutants, compared to matched sibs (Fig. 7c–f; Supplementary Fig. 6b). The total number of fibres in single body cross-sections of a mutant (4097) and a heterozygote sib (3657) were similar and were approximately 40-fold those in larvae, indicating that fibre formation had persisted (Supplementary Fig. 6b). The number of nuclei in mutant fibre cross-sections was reduced threefold compared to sibs, but a few fibres with several nuclei were still present (Fig. 7g, h; Supplementary Fig. 6d, e). The total area of both slow and fast muscle also appeared reduced (Fig. 7i; Supplementary Fig. 6b). Fibre typing for oxidative metabolism revealed that fast muscle fibre size was more reduced than slow fibre size. Interestingly, the slow muscle region overall appeared less oxidative and fibres in the intermediate and slow regions were disorganised and smaller in mutant fish (Fig. 7i). Thus, fish lacking Myogenin have severe defects in adult muscle.

**Fusion defect in adult-derived *myog* mutant MPCs.** To determine whether the deficits in mutant adults derived from defective development alone, or persisted due to a Myog requirement in the adult, we analysed cellular dynamics of adult-derived muscle progenitors cells (MPCs) ex vivo. Initially, we developed a method to culture satellite cells derived from isolated fibres of adult zebrafish that yielded MPCs that could undergo terminal differentiation, up-regulate desmin and accumulate MyHC and fuse into myotubes in vitro (Fig. 8a). Strikingly, *myog* mutant MPCs differentiated as well as sibling MPCs (Fig. 8b), but showed a dramatic reduction in cell fusion compared to heterozygote controls (Fig. 8a–d). Indeed, fusion index declined greatly; whereas the majority of sibling myotubes contained three or more nuclei, 98% of mutant myotubes were mononucleate (Fig. 8c, d, Supplementary Fig. 6f). Differentiating Myog-deficient MPCs expressed desmin and MyHC, elongated and aligned similarly to control cells (Fig. 8a, b; Supplementary Fig. 6g). These data show Myog is not required for terminal differentiation of satellite cell-derived MPCs into myocytes but is required for myocyte fusion throughout life.

## Discussion

The data presented radically change the interpretation of the evolutionarily conserved role of Myog in skeletal muscle development through four major findings. First, Myog is not required for terminal differentiation of most myoblasts into myocytes. Second, Myog has a major conserved role in driving the fusion of myocytes into multinucleate fibres. Third, that a second, Myog-independent, pathway to muscle fusion exists and, in the zebrafish trunk, is promoted by Hh signalling. Lastly, that Myog is required for normal myogenesis throughout life and that its loss leads to poor muscle and whole body growth and a persistent

functional fusion deficit in adult satellite cell-derived muscle progenitors.

In mice lacking Myog, myoblasts can form myocytes expressing proteins of the contractile apparatus[5–8]. However, a major deficit of early muscle formation was reported, with dramatic downregulation of MyHC at e12.5 in both trunk and limb muscle and a worsening deficit in neonates[7,8]. This led to the view[2], that 'Myogenin knockout has a …. complete absence of functional skeletal muscle'. Our data from zebrafish contradict this view; we observe differentiated muscle fibres, normal sarcomere formation and normal numbers of nuclei within fibres in the larval myotome of mutants. Nevertheless, although to our knowledge no compelling images of fusion in the absence of Myog in vivo have been published, cultured myoblasts and satellite cells from *Myog* mutant mice are reported to fuse and myocytes lacking Myog can fuse with wild-type myocytes in vivo in murine chimaeras[7,11]. We conclude that Myog is dispensable for terminal differentiation of myoblasts into post-mitotic myocytes and contractile myofibres in zebrafish (Fig. 8e).

Early reports suggested that murine myotomal Myog protein does not accumulate until after muscle differentiation, despite the earlier presence of *Myog* mRNA[56,57]. Moreover, embryonic MyHC (*Myh3*) mRNA is almost normally expressed in e14.5 *Myog* mutants, whereas maturation to expression of perinatal MyHC (*Myh8*) mRNA is dramatically reduced at e18.5[7,8]. The widespread accumulation of desmin protein, hitherto taken as an indication of a myoblast state[7,8,58], could instead reflect myocyte formation. The ability of myogenic cells from *Myog*[−/−] to form myotubes in culture also argues for unimpaired myocyte formation[6,7]. Consistent with murine data[8], we find that balance of zebrafish slow and fast fibre formation is unaffected by loss of Myog. However, our finding that several genes required for later steps in myocyte differentiation (e.g. fusion) are down-regulated in mutants indicates that a subset of muscle differentiation genes, the 'Myog-module' are regulated by Myog in zebrafish (Fig. 8e).

Zebrafish Myog is not essential for, but can promote, muscle terminal differentiation. We previously reported that loss of both Myf5 and Myod ablates all skeletal myogenesis, whereas combined reduction of Myod and Myog severely reduces fast muscle[17,18]. These findings show that, in the absence of MyoD, Myf5 requires Myog to drive fast myogenesis[17]. Interestingly, even hypomorphic *myog*[fh265] mutants that lack a developmental fusion defect show poor muscle regeneration and apoptosis of *myf5*-marked cells[20]. Hence, we cannot exclude the possibility that a non-essential subset of myoblasts requires Myog for terminal differentiation in older zebrafish. Lack of Myog is not compensated by increased expression of *myod* and leads to downregulation of *mrf4* mRNA, as observed in mice lacking Myog[5,7,8,12]. Surprisingly, we found that *myog* mutant embryos accumulate less *myf5* mRNA, suggesting that Myog may promote *myf5* expression or be required for the production of *myf5*-expressing cells.

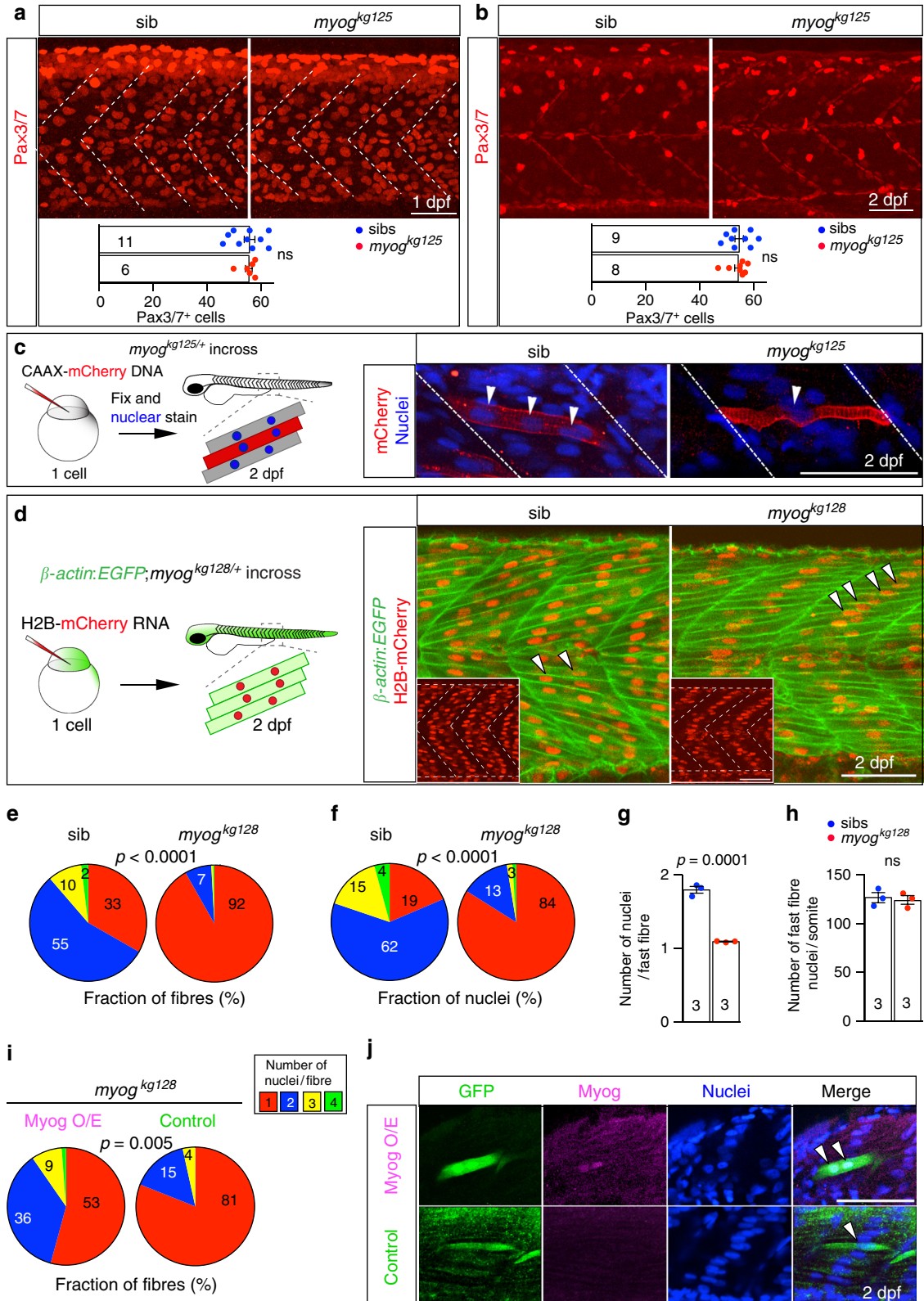

*Myog* mutants have a severe lack of fusion, despite efficient myocyte differentiation. We find that *myog* activity is essential for normal expression of a subset of fusogenic genes, *mymk*, *mymx* and *jam3b*, mutation of which causes fusion defects strikingly similar to the *myog* mutant phenotype[23–25,48,49,59]. We propose, therefore, that early terminally-differentiated fast myocytes are primed for fusion, but lack sufficient expression of critical Myog-module components until Myog becomes active (Fig. 8e). Interestingly, *mymk* mRNA was more highly reduced than any other gene analysed, which could account for the fusion defect observed. However, as *jam3b* mRNA is more widely expressed[48,60], its lesser reduction may reflect strong reduction in

**Fig. 4** Myogenin promotes fusion of myocytes. **a, b** Immunodetection and quantification of Pax3/7 positive MPCs in somite 17 of $myog^{kg125/+}$ incross embryos at 1 dpf and 2 dpf. Mean ± SEM of dots representing individual embryos. Lack of Myog does not alter number of Pax3/7 MPCs per somite (white dashed lines). Representative images $n = 6$ mutants, $n = 11$ sibs (1 dpf); $n = 8$ mutants, $n = 9$ sibs (2 dpf). **c** Qualitative analysis of myoblast fusion in a $myog^{kg125/+}$ incross injected at 1-cell stage with DNA encoding CAAX-membrane targeted mCherry (red). At 2 dpf, larvae were fixed and stained with Hoechst to highlight nuclei (blue) and analysed from 3D stacks. Nuclei within mCherry-labelled fibres (arrowheads) were mostly single in mutants, but multiple in sibs. Representative images $n = 6$ embryos. **d** Myoblast fusion quantified by injection of $H2B:mCherry$ RNA into 1-cell stage embryos from $\beta$-$actin:EGFP;myog^{kg128/+}$ incross. Confocal single plane images deep in the myotome of 2 dpf larvae showing muscle fibres and the position of nuclei (insets). Note the central location away from somite borders (dashed white lines) of most nuclei in mutants (arrowheads), similar to that observed in mononucleate superficial slow fibres. Representative images $n = 6$ embryos. **e, f** Quantification of fusion within the entire myotome 17, showing the fraction of fast fibres (**e**) and fraction of nuclei in fast fibres (**f**) with the indicated number of nuclei. Slow fibre numbers were unaltered. Data report mean values of three larvae per genotype (see Supplementary Fig. 3a for individual data). $p$-values indicate probability of rejecting null hypothesis of no difference between mutant and sibs in $\chi^2$ tests. **g** Number of nuclei per fast fibre is reduced in mutant. **h** Total number of nuclei within fast fibres in somite 17 of sib and mutant is unchanged. Dots represent individual embryos. Mean ± SEM. $t$ test. Bars = 50 μm. **i, j** Mosaic $myog:MyogCDS-IRES-GFP$ plasmid-derived expression of Myog (Myog O/E) rescues fusion in $myog^{kg128}$ mutant larvae from a $myog^{kg128/+}$ incross, compared control $myog:GFP$ plasmid (Control). Quantification (as in **e**) of nuclei in GFP$^+$ cells (**i**, see Supplementary Fig. 4a for individual data). Immunodetection shows Myog overexpression (Myog O/E, magenta) in $myog:MyogCDS-IRES-GFP$ but not in $myog:GFP$ (Control, green) GFP$^+$ fibres (**j**). Representative images $n = 5$ $myog:GFP$, $n = 11$ $myog:MyogCDS-IRES-GFP$ injected mutants, ns: not significant

muscle and unaltered expression elsewhere. Thus, the extent of the Myog-module functionally required for fusion remains to be determined.

Mosaic Myog re-expression in $myog$ mutant fully rescues fusion. Strikingly, this effect is cell autonomous and fairly efficient, which leads to several important conclusions. Firstly, as two adjacent GFP-marked cells are rare, a single isolated Myog-expressing myocyte appears sufficient to induce fusion to an adjacent cell lacking Myog. This result parallels the chimaera analysis showing fusion of $Myog^{-/-}$ with wild-type cells in murine myogenesis[11]. Secondly, the existence of rescued fibres with more than two nuclei suggests that Myog expression in a binucleate fibre can elicit fusion of adjacent cells lacking Myog. It may be significant in this regard that Jam3b, Mymk and Minion/Myomerger/Myomixer have been shown to be required in only one cell of a fusing pair, although fusion efficiency was reportedly higher when both cells express Mymk[24,47,48].

We observed that Myog occupies E-boxes in the endogenous $mymk$ 5' proximal region during the period of fusion in vivo. A 3 kb $mymk$ promoter fragment containing these sites drives reporter expression in zebrafish fast muscle[24]. Similarly, in mouse and chick, $Mymk$ expression parallels that of $Myog$ and depends on conserved E-boxes, including one at -41 bp[61,62]. Moreover, in cultured myocytes, Myog binds to conserved sequences in $Mymk$, $Mymx$ and $Jam3$ (www.encodeproject.org/experiments/ENCSR000AID). Interestingly, the early Xenopus, chick and mouse myotomes are reported to be composed of mononucleate fibres[63–67]. As the murine $Myog$ mutant shows little early defect, but even partial deletion of a floxed $Myog$ allele after initial fibre formation shows that Myog is essential for late embryonic and neonatal myogenesis[12,13], the data on murine $Myog$ mutants are all consistent with a primary fusion defect. Moreover, some reports in C2C12 cells have suggested that $Myog$ expression correlates with, and is required for, myocyte fusion, although conflicting findings exist[68,69]. We note that zebrafish slow muscle fibres, which remain mononucleate long after their terminal differentiation, accumulate much lower levels of $myog$ mRNA than fast muscle precursors, paralleling their lower levels of $mymk$ and $mymx$ mRNA[17–19,59,70]. However, although overexpression of Myog in slow myocytes reduces their size and alters myofibril organisation, it did not drive their fusion to each other or to adjacent fast myocytes, in contrast to $mymk$ overexpression[24].

Zebrafish $myog$ mutants show a similar increase in number of myofibres to that reported when fusion is blocked in mutants lacking Jam proteins[48]. In the case of $myog$ mutants, the increase in fibres was quantitative; total nuclear number in myofibres

remained constant, suggesting that normal numbers of myoblasts differentiated, survived and made fibres. It seems that during early myotome formation, therefore, either no specific sub-population of 'founder' myoblasts determines fibre number, as occurs in Drosophila embryonic myogenesis[71,72] or Myog is required to prevent cells acting as founder cells. Interestingly, previous studies reported that fusion-defective myoblasts elongate and differentiate into mononucleated muscle fibres[23,24,48,49]. We hypothesise that, like fish, murine $Myog$ mutants are blocked in fusion. Perhaps lack of fusion triggers loss of nascent myocytes that fail to form attachments to skeleton or nerve. In addition, myoblast populations such as those in neonatal limb may absolutely require Myog for terminal differentiation.

In $myog$ mutant fish a small group of muscle fibres in the deep myotome undergoes fusion. Residual fusion was also observed in $jam3b$ and $jam2a$ mutants and $mymk$ mutants were also stated to be only 'predominantly mononucleated', although the extent and location of residual multinucleate fibres was not reported[24,48]. We find that Myog-independent fusion requires Hh signalling, probably from adjacent midline tissue, which also up-regulates $mymk$ mRNA in the deep myotome. The low residual level of $mymk$ mRNA observed in CyA-treated $myog$ mutants could reflect incomplete loss of Hh function, or additional controls on $mymk$ expression. As Hh promotes slow and fast muscle differentiation through activation of Myod by a Cdkn1c/p57 positive feedback loop[19], we speculate that Hh-induced fusion may arise from increase in Myod-driven $mymk$ expression in myocytes adjacent to the midline source of Hh. Our data suggest that distinct myocyte fusion processes contribute to muscle fibre diversity.

Our study quantified the extent of fusion in wild-type and $myog$ mutant fast muscle. Interestingly, a significant minority (~30%) of fast fibres are mononucleated at 2 dpf in wild-type, meaning that ~20% of fast myocytes had not yet fused. Many of these may reflect dermomyotome-derived fast myocytes that had recently undergone terminal differentiation, because they were predominantly located in the lateral myotome[36,38]. Nevertheless, transplant experiments have reported rates of fusion above 95%[24,48], implying that a rare subset of proliferative somite cells generates the mononucleate fast fibres. On the other hand, in $myog$ mutants only about 20% of myocytes fuse. As frequencies of residual fusion are yet to be reported in $jam2a$, $jam3b$, $mymk$ and $mymx$ mutants[23,24,48,59], it is unclear whether some fusion mutants have more severe defects than others.

$Myog$ mutants are viable but grow less rapidly than their sibs. Early in life, $myog$ mutants have small fibres and reduced myotome size, which might give them a disadvantage in competitive

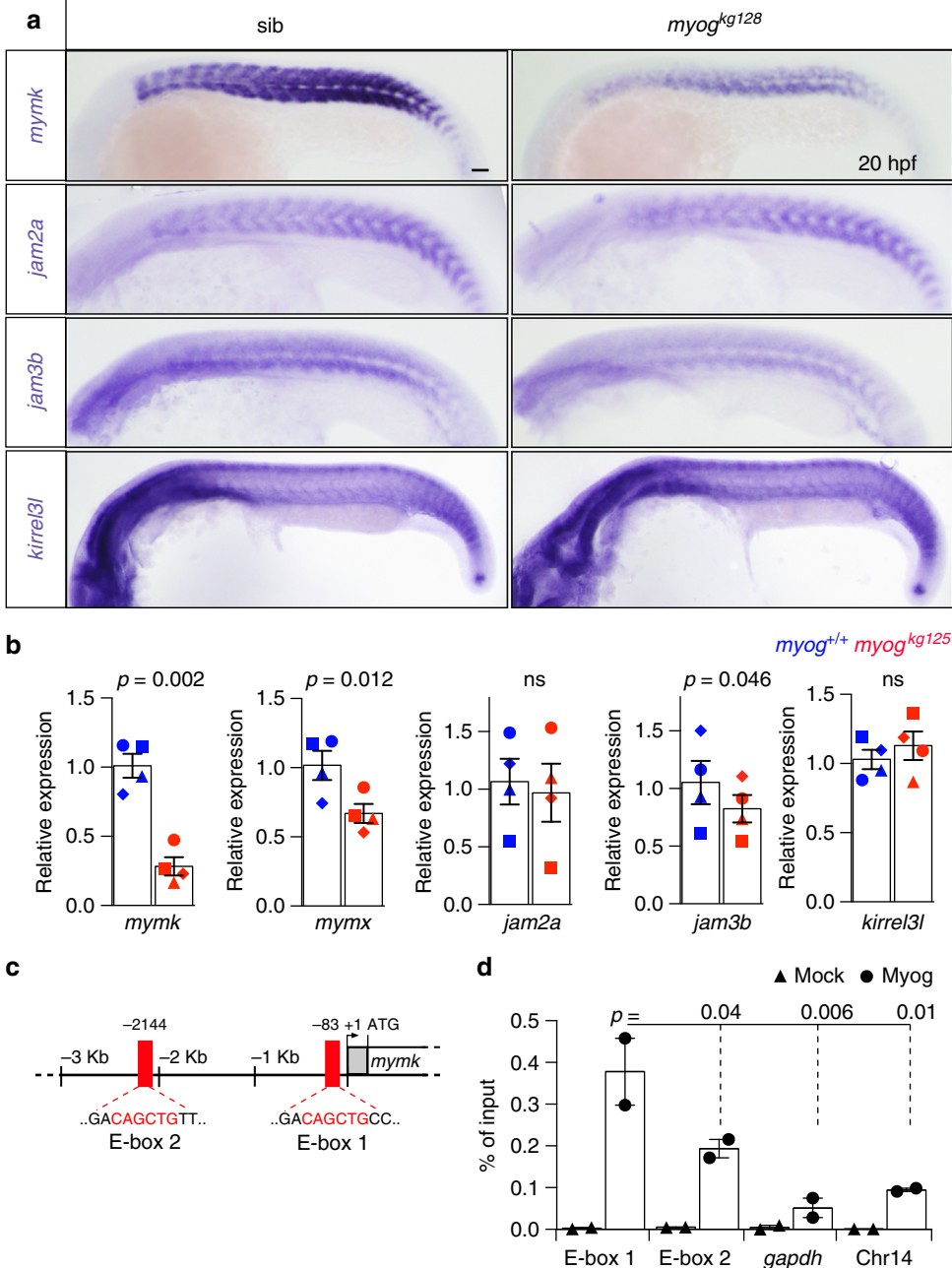

**Fig. 5** Myogenin mutants have reduced expression of fusogenic factors. **a** ISH on 20 hpf *myog*^kg128/+ incross to investigate the expression of genes essential for vertebrate myocyte fusion. Expression of *myomaker* (*mymk*) and *jam3b*, but not of *jam2a* or *kirrel3l*, is reduced in *myog* mutant. Bar = 50 µm. Representative images *n* = 10 mutants, *n* = 38 sibs (*mymk*); *n* = 13 mutants, *n* = 37 sibs (*jam2a*); *n* = 12 mutants, *n* = 39 sibs (*jam3b*); *n* = 7 mutants, *n* = 13 sibs (*kirrel3l*). **b** qPCR analysis of RNA expression levels showing downregulation of *mymk*, *myomixer/myomerger/minion* (*mymx*) and *jam3b* at 20 hpf on *myog*^kg125 mutants, whereas *jam2a* and *kirrel3l* remain unaltered compared to wt (*myog*^+/+) sibs. Graphs show mean fold change ± SEM of four independent experiments; paired *t* test. Symbol shapes denote wt and mutant samples from paired experiments. **c** Schematic of 5' genomic region of *mymk* reporting: 5'-UTR and coding sequence (grey and white boxes), position of E-box elements (red boxes), relative bp distance from 5'UTR start (+1), start codon (arrow, ATG). E-box 1 and E-box 2 sequences are shown in red text. **d** ChIP-qPCR assay using anti-Myog or mock showing significant Myogenin enrichment on *mymk* E-box 1 compared to negative controls from the *gapdh* promoter and a gene-free region on chromosome 14 containing an E-box (Chr14). Mean of percentage of input immunoprecipitated ± SD of two independent experiments, ANOVA

feeding leading to reduced growth. Alternatively, Myog could be required for some other function, such as synthesis of myokines important to coordinate whole body scaling of tissue size. However, adult mutants have a disproportionate loss of muscle compared to their length and a persistent greatly reduced fibre size and nuclear content throughout life that could reflect defective adult MPC differentiation.

Zebrafish muscle has been shown to contain fibre-associated satellite cells[41,73,74]. To address the role of Myog in adult life, we developed an MPC culture procedure for zebrafish satellite cells. This method shows that adult MPCs lacking Myog undergo terminal differentiation but fuse poorly ex vivo, indicating that Myog is required throughout life, rather than that the defect in adult fish derives solely from persistence of early developmental defects

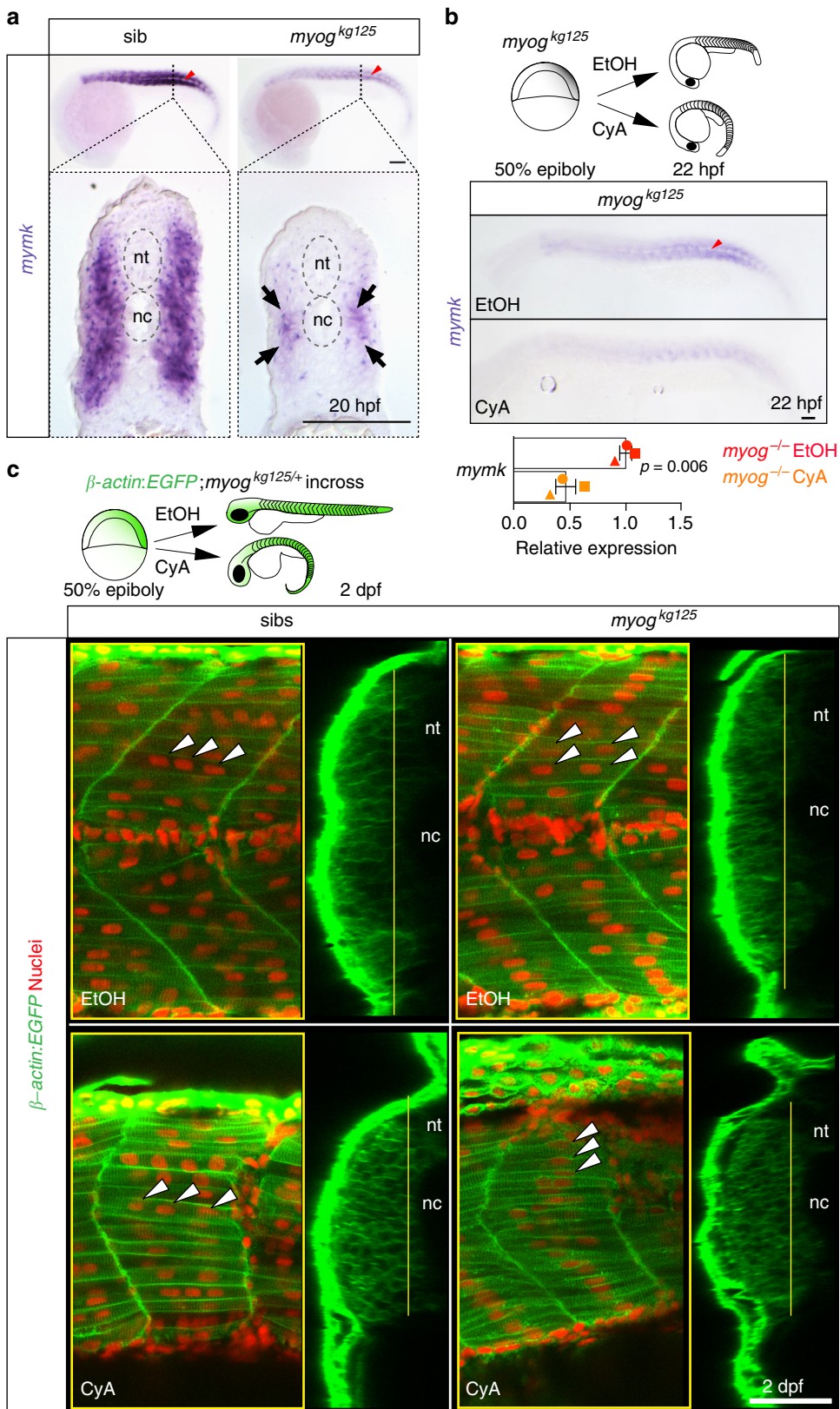

that are subsequently Myog-independent. Definitive proof of this conclusion will require deletion of *myog* function in adult fish. Nevertheless, combined with the defective regeneration of larval muscle in *myog* hypermorphs[20], our data strongly suggest that Myog also functions during adult muscle growth and regeneration.

In striking contrast to the *myog* mutant phenotype, lack of Myod reduces fibre number in larvae without affecting fusion or *mymk* expression, and the remaining fibres grow larger[45]. In *myod* mutants, a reduction in the number of fast fibres is accompanied by an increase in MPCs expressing Pax3 and/or

**Fig. 6** Hedgehog signalling sustains residual fusion and *mymk* expression. **a** ISH for myomaker (*mymk*) at 20 hpf revealed that residual expression in *myog* mutant is enriched in the medial region of the somite close to notochord (arrows in transverse sections from indicated axial level, dorsal to top). Note lack of expression in mononucleate slow pioneer fibres (arrowheads, upper panel). Representative images $n = 6$ mutants, $n = 14$ wt sibs (*mymk*) **b** ISH (lateral view, dorsal to top) and qPCR analysis showing that cyclopamine (CyA) treatment of $myog^{kg125}$ embryos almost abolished *mymk* mRNA compared to ethanol (EtOH) vehicle control. CyA effectiveness is shown by the absence of unstained slow muscle pioneer cells (arrowhead). Mean fold change ± SEM from three independent experiments on embryos from separate lays of $myog^{kg125}$ (circles) and $myog^{kg128}$ (squares and triangles) analysed on separate days, paired $t$ test statistic. Representative images $n = 4$ EtOH, $n = 6$ CyA. **c** Optical confocal sections of the medial region of somites 17 of $\beta$-actin:EGFP; $myog^{kg128/+}$ incross treated with vehicle or CyA. Transverse-section panels show medial position (yellow lines) of respective longitudinal section for each condition. CyA abolished residual fusion in the medial myotome of mutant embryos (arrowheads) but did not detectably affect fusion in sibs. Note that the residual multinucleate fibres in $myog^{kg125}$ mutant appear larger than adjacent mononucleate fibres in EtOH but are lacking in CyA. nt: neural tube, nc: notochord. Representative images $n = 5$ mutants, $n = 3$ sibs (EtOH); $n = 4$ mutants, $n = 6$ sibs (CyA). Bars = 50 μm

Pax7, presumably reflecting reduced terminal differentiation[18,45]. Given that Myod can activate *myog* expression and either MRF can drive terminal differentiation of fast fibres[17], these opposite effects on fibre number and size suggest that the balance of MRFs influences the mode of muscle growth.

Several studies in mice have also implicated MRFs in adult fibre size control[14,75,76]. Like zebrafish, mice conditionally lacking Myog in the adult show reduced body and fibre size, although nucleation state of mutant myofibres was not reported[12–14]. However, in contrast to its role in promoting fusion and fibre growth in developing muscle, in the adult context Myog appears to promote fibre atrophy upon denervation and regulate metabolic capacity in innervated muscle[14,77]. Moreover, Myog overexpression in both multinucleate fast muscle in mice and, as shown here, in mononucleate slow fibres in larval zebrafish leads to reduced fibre diameter[75]. As suggested by the differences in Myog-regulated genes in embryonic and adult cells[13], it seems that Myog performs distinct functions in myoblasts, nascent myocytes and mature fibres.

## Methods

**Zebrafish lines and maintenance**. All lines used were reared at King's College London on a 14/10 h light/dark cycle at 28.5 °C with adults kept at 26.5 °C, with staging and husbandry as described[78]. Embryos/larvae were reared at 28.5 °C in the dark, except for periods outside the incubator. $myog^{fh265}$ and $myod^{fh261}$ mutant alleles[18,45] on AB background were genotyped by sequencing as described previously[18]. $myog^{kg125}$ and $myog^{kg128}$, on the TL background, were genotyped by sequencing or by loss of EcoRV site in the mutant alleles, following PCR amplification using primers indicated (Supplementary Table 1). The two new alleles had indistinguishable phenotypes and no differences were detected between wt and heterozygous fish, so we refer to mutants and siblings (sibs), and report the specific *myog* allele in each experiment in Figures and Supplementary Table 2. *Tg(Ola.Actb: Hsa.HRAS-EGFP)$^{vu119}$* (ref. [33]) was originally on King's wild-type background. All experiments were performed on zebrafish derived from F2 or later filial generations, in accordance with licences held under the UK Animals (Scientific Procedures) Act 1986 and later modifications and conforming to all relevant guidelines and regulations.

**Embryo manipulation**. *Myogenin* mutants were generated targeting the sequence 5′-GGAGCTCCTGTCCTGATATC-3′ on the reverse strand using CRISPR/Cas9 method as previously described[79]. Mutant lines, $myog^{kg125}$ and $myog^{kg128}$, were bred onto *Tg(Ola.Actb:Hsa.HRAS-EGFP)$^{vu119}$*. Muscle (myotome) size was analysed as previously described[18,45] and schematised in Fig. 3d, except for $myog^{fh265/+}$ incross embryos, which were immersed in 3 μM BODIPY-FL-C5 (Thermo Fisher Scientific) in fish water (FW) from 30 hpf until 2 dpf, washed twice and agarose-mounted for live imaging[18,45]. H2B-mCherry capped RNA (100 pg per embryo, kind gift from H. Roehl, University of Sheffield, UK and S. Megason, Harvard Medical School, USA) or DNA plasmid encoding membrane targeted CAAX-mCherry (25 pg per embryo)[80] were injected into 1- to 2-cell stage embryos to analyse fusion. Morpholino antisense oligonucleotide against $myog$[17,18] (2 ng per embryo) was injected into 1- to 2-cell stage *Tg(Ola.Actb:Hsa.HRAS-EGFP)$^{vu119}$* embryos, which were fixed with 4% paraformaldehyde (PFA) for 15 min, at 20 ss when knockdown efficiency was checked by Myog immunodetection, or at 2 dpf and incubated overnight with Hoechst 33342 (Life Technologies). Cyclopamine[81] (50 μM) or ethanol vehicle control was added at 50% epiboly to embryos with chorions punctured with fine forceps.

Motor function was assayed at 5 dpf in a 30 min trial using DanioVision (Noldus) and EthoVision XT9 tracking software. Larvae were acclimatised in 0.6% methyl-cellulose (MC, Sigma Aldrich) or FW vehicle control in 24-well plates for at least 2 h before tracking. Following tracking, larvae were raised in MC or FW and analysed by confocal imaging at 8 dpf.

**Rescue assay**. *Myogenin* coding sequence was PCR amplified from pBluescript SK-MG12-ZF-*Myogenin*[70] using listed primers (Supplementary Table 1) and subsequently cloned into *hsp70-4:MyogCDS-IRES-NLSmGFP6*[82]. *MyogCDS-IRES-NLSmGFP6* insert was then PCR amplified from *hsp70-4:Myog-IRES-NLSmGFP6*. *myog promoter:GFP* vector (*myog:GFP*)[83] was linearised by PCR removing GFP sequence using listed primers (Supplementary Table 1). Final *myog:MyogCDS-IRES-GFP* plasmid was made using Gibson Assembly (E2621, NEB) and sequence verified (Genbank:MH593821). Rescue experiments were performed by injecting 20 pg *myog:MyogCDS-IRES-GFP* or *myog:GFP* control into 1- to 2-cell stage $myog^{kg128/+}$ incross lays. Embryos were fixed at 2 dpf and processed for immunostaining for Myog, GFP, slow MyHC (F59) and Hoechst 33342 as described below. Each embryo was then mounted for confocal scanning of somites 15-20 on one side and nuclei within GFP$^+$ fibres were counted.

**Imaging and in situ mRNA hybridisation and immunodetection**. ISH and immunodetection were performed as described[84]. Briefly, fish were fixed in 4% PFA in phosphate-buffered saline (PBS) for 30 min or 3 h at room temperature or overnight at 4 °C. Embryos for ISH were stored in 100 % methanol at −20 °C and rehydrated in PBS prior to ISH. Fish for immunostaining were permeabilised in PBS 0.5% Triton X-100 (PBSTx) for 5 min, blocked in Goat Serum 5% (Sigma Aldrich) in PBSTx and incubated with primary antibodies at indicated concentrations at least overnight. Fish were then washed in numerous changes of PBSTx for at least 5 min and incubated and washed similarly with indicated secondary antibodies and prepared for imaging as described below. Primary antibodies against Myog (M-225 Santa Cruz Biotechnology, 1:50), fast MyHC (EB165 (1:2), Developmental Studies Hybridoma Bank, Iowa (DSHB)), slow MyHC (S58 (1:2) or F59 (1:5), DSHB) or MyHC (A4.1025 (1:5)[30], MF20 (1:300, DSHB)), α-actinin (1:500, A7732, Sigma Aldrich), Pax3/7 (DP312 (1:50), Nipam Patel, UC Berkeley, USA), Laminin (L9393 (1:400), Sigma Aldrich), GFP (13970 (1:400), Abcam), Titin (T12 (1:10), D. Fürst, University of Bonn, Germany), desmin (D8281 (1:100), Sigma Aldrich) were detected with Alexa-conjugated secondary antibodies (Invitrogen) and Goat anti-Mouse IgA-FITC (Serotec). Digoxigenin-labelled probes were against $myog$[70], $smyhc1$[27], $mylpfa$[26] or *mymk, jam3b, jam2a* and *kirrel3l* made by PCR on 1 dpf cDNA template using listed primer pairs (Supplementary Table 1) with an added T7 polymerase binding site. For confocal imaging, embryos were mounted in glycerol, Citifluor (Agar) or 0.8–1% low melting point agarose and data collected on the somites 17-18 near the anal vent on a LSM Exciter microscope (Zeiss) equipped with 20 × /1.0 W objective and subsequently processed using either Volocity (Perkin Elmer), Fiji (NIH, www.Fiji.sc) or ZEN (Zeiss) software. Myotome volume, number of fibres and fibre volume were calculated as described[45] and schematised in Fig. 3d. α-Bungarotoxin-Alexa 488 (Invitrogen) and phalloidin staining was as described[85]. Myogenin immunofluorescent intensity was averaged from at least 10 randomly-selected nuclei at similar dorso-ventral and mediolateral position within somites 9-10 of each embryo. Nuclei of interest (NOI) were selected blind by Hoechst 33342 staining then Myogenin fluorescence was quantified in selected NOI using Fiji. To account for staining variability between embryos, background was subtracted from nuclear Myogenin intensity in each embryo by measuring fluorescence intensity of nucleus-free areas of an equal size to the NOI in each somite region (see Supplementary Fig. 1a for schematic). Relative average fluorescence intensity of nuclear Myogenin immunolabelling in mutant was then calculated relative to siblings. All images are shown as lateral views with anterior to left and dorsal up, unless otherwise stated.

**RT-PCR and qPCR**. $myog^{kg125/+}$ incross embryos at 20 hpf were individually genotyped by sequencing using listed primers (Supplementary Table 1). RNA was

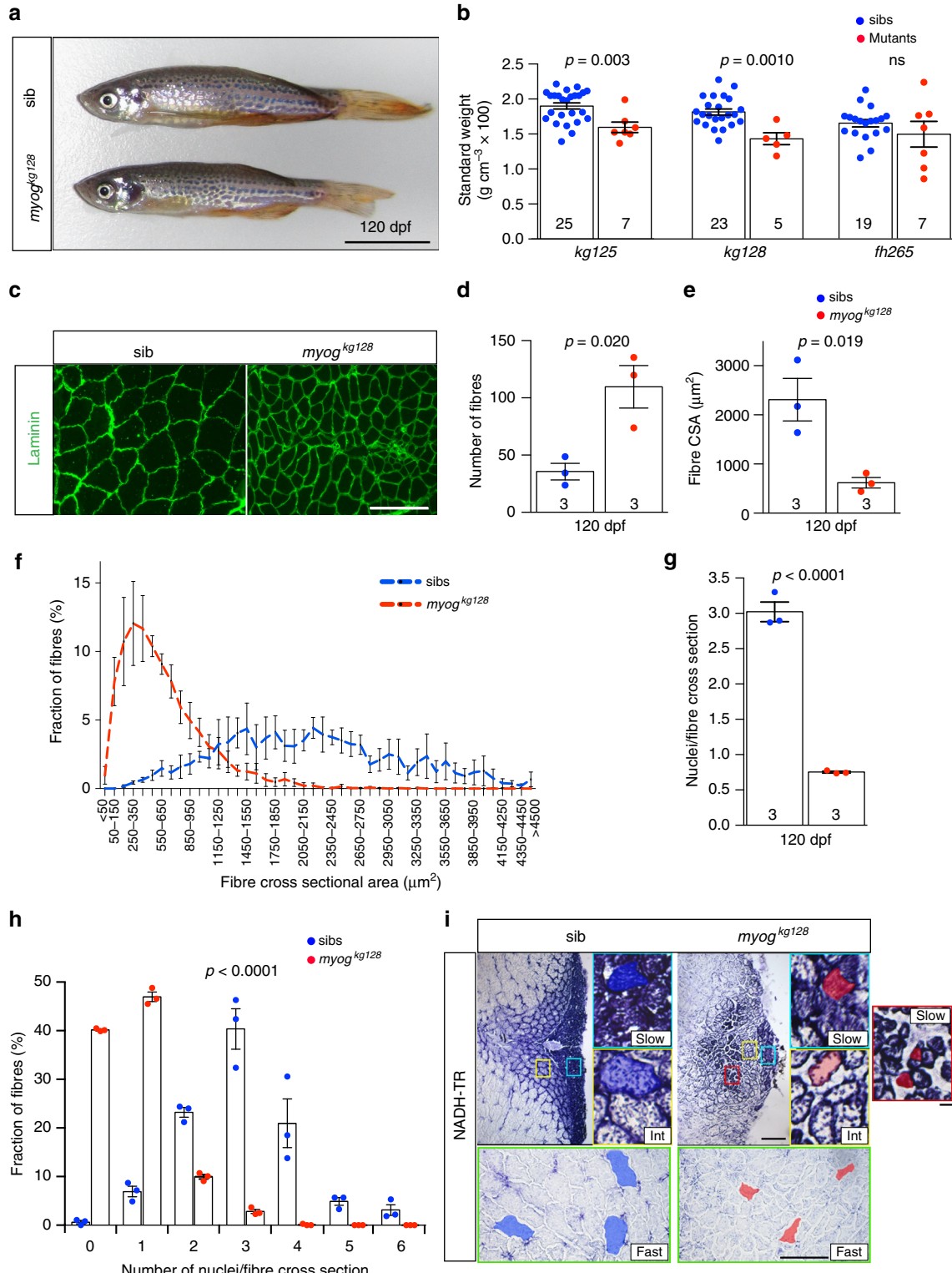

extracted from pools of four embryos of each genotype using Trizol® (Sigma Aldrich) and purified with RNA Clean & Concentrator™-25 (Zymo Research) or RNA Purification Plus Kit (Norgen). Total RNA (300 ng) was reverse transcribed using Superscript III reverse transcriptase (Invitrogen) following supplier's instructions. qPCR on technical triplicates for each sample was performed on 5 ng of relative RNA using Takyon Low ROX SYBR 2X MasterMix blue dTTP (Takyon) on a ViiA™7 thermal cycler (Applied Biosystems). For each experimental sample, $\Delta CT$ was calculated by subtracting the CT value for housekeeping gene (actb2) from that of the target gene. $\Delta\Delta CT$ of each target gene was then calculated by subtracting the average of the $\Delta CT$ obtained in the wt (sibilngs) samples from $\Delta CT$ for each sample. Relative gene expression was calculated using the $2^{-\Delta\Delta CT}$

formula[86] and the fold change of the expression levels between sibs and mutants were compared using paired Student's t test. Results are presented as mean ± SEM of fold changes from three or four independent experiments. Primers were purchased from Sigma-Aldrich (KiCqStart® SYBR® Green Primers Predesigned, Sigma Aldrich). All PCRs for genotyping and probe synthesis were performed using Phusion Taq polymerase (Life Technologies) on a T100 thermal cycler (Bio-Rad).

**Chromatin Immunoprecipitation and E-box enrichment analysis**. 3 Kb of putative promoter region of *myomaker*, retrieved from UCSC genome browser (GRCz10/danRer10), was scanned for E-box elements using JASPAR 2016 version

**Fig. 7** Adult Myogenin mutants have reduced muscle with more but smaller myofibres. **a** *Myog* mutant and sib at 120 dpf from *myog*[kg128/+] incross. Bar = 1 cm. Representative images n = 5 mutants, n = 23 sibs. **b** *Myog*[kg128] or *myog*[kg125] but not *myog*[ffh265] showed reduced standard weight compared to co-reared sibs at 120 dpf. Dots represent individuals. **c** Laminin immunodetection on cryosections from 120 dpf *myog*[128/+] incross. Bar = 100 μm. Representative images, n = 3. **d**–**f** Number of muscle fibres in 0.1 mm² of adult muscle is increased in mutants (**d**), whereas myofibre cross-sectional area (CSA) is decreased (**e**) reflecting a shift in CSA frequency distribution compared to sibs. **g** Fewer myonuclear profiles were present within laminin profiles in adult muscle cross-sections in mutants than in sibs, measured from 107 to 490 fibres at similar medio-lateral and dorso-ventral positions of trunk muscle of three fish per genotype. Mean ± SEM, *t* test. **h** Proportions of muscle fibres with indicated number of myonuclei within fibre cross-sectional profile. In sibs, >90% of fibres have more than one nuclear profile, compared with <15% in mutants. Mean ± SEM, $\chi^2$ test. **i** NADH tetrazolium reductase stain revealed that in both mutants and sibs three fibre types are present: oxidative/slow (slow), intermediate (int) and glycolytic/fast (fast). Size of more glycolytic myofibres (yellow and green insets) is more reduced than oxidative fibres (cyan). Assay was performed on three 120 dpf adult male length-matched fish of each genotype. Representative sib (blue) or mut (red) fibres are highlighted. Mutant presents smaller slow type myofibres ectopically localised in fast domain (red inset). Representative images, n = 3. Bars = 100 μm (except for red, yellow and cyan insets = 10 μm)

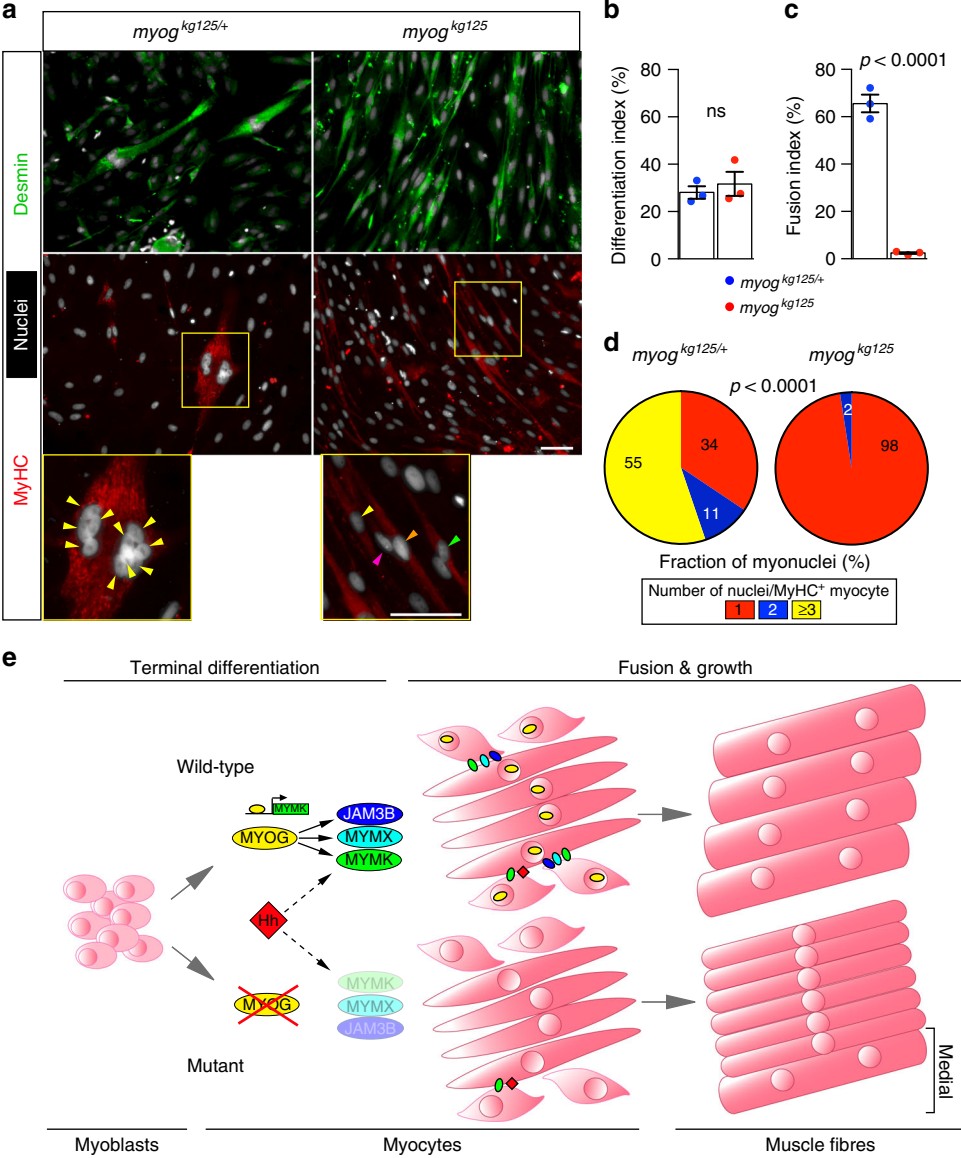

**Fig. 8** Mutant adult-derived muscle progenitor cells retain fusion deficit ex vivo. **a** Immunodetection of desmin (green) and MyHC (red) and nuclei (white, Hoechst) in 15 months old *myog*[kg125] and sibling *myog*[kg125/+] adult-derived MPCs following 5 days of differentiation. Fusion into multinucleated myofibres occurred only in sib (magnified boxes), coloured arrowheads indicate nuclei of each cell. Representative images, n = 3. **b** Extent of differentiation (Differentiation index) is comparable between mutant and heterozygous MPCs. **c** Fusion index showing deficit in fusion of mutant myocytes. **d** Number of nuclei in fused MyHC⁺ cells is reduced in mutant, $\chi^2$ test. Three fish per genotype (three technical replicates each). **e** Schematic of the role of Myogenin during differentiation, fusion and growth of muscle fibres. During myogenesis, committed MPCs leave the cell cycle, begin to elongate and express early muscle-specific genes during terminal differentiation into myocytes. At this stage, Myogenin (MYOG) promotes the expression of *myomaker* (MYMK), *myomixer* (MYMX) and *jam3b* (JAM3B). These fusogenic proteins prompt myocyte fusion to form muscle fibres. In the absence of Myogenin, myocytes undergo terminal differentiation but fail to express Myog-module genes, remain mononucleated and grow less throughout life. Residual myocyte fusion in *Myog* mutants in the medial region of the somite (bracket) is sustained by Hedgehog (Hh) signalling

(JASPAR CORE Vertebrata, jaspar.genereg.net) with default settings. To avoid false positive scoring, relative profile score threshold was set at 95-99%. 500–600 20 hpf wt TL embryos were processed as described[87]. For immunoprecipitation, 30 μg of chromatin were incubated with 4 μg of Myogenin antibody or normal rabbit serum as mock. After purification of the immunoprecipitated DNA, enrichment was analysed by qPCR using primers listed in Supplementary Table 1. Control primer pairs for genomic *gapdh* sequence and for chromosome 14 gene-free region were described[87,88]. All signals were normalised for input by percentage input calculation method (www.thermofisher.com).

**Adult fish analysis**. Siblings (120 dpf or 15 mpf) from heterozygote incrosses were anaesthetised with tricaine (Sigma Aldrich), blotted dry, weighed on an Ohaus YA102 balance, nose-to-base of tailfin length measured with a ruler and fin-clipped for sequence genotyping. Standard weight (K) was calculated using Fulton's formula $K = weight (g) \times 100 \times length^{-3} (cm)$ (reviewed in ref.[89]). Body mass index (BMI) was calculated as 'weight (g) × length$^{-2}$ (cm)'. Three 120 dpf adult male length-matched fish of each genotype were culled with high dose tricaine, eviscerated and skinned. Trunk from just behind gills to 5 mm beyond the dorsal fin was embedded in OCT (CellPath, Fisher Scientific), immersed in freezing isopentane (Fisher Scientific) and stored at −80 °C. Cryosections (15 μm) from three anteroposterior positions were immunolabelled for Laminin and counterstained with Hoechst 33342 as described[90] and three or four images in consistent mediolateral and dorso-ventral somitic areas of sibs and mutants were acquired using an Axiovert 200 M microscope (Zeiss) equipped with LD A-plan ×20/0.85 objective. Fibre cross-sectional area (CSA) was measured in each image and averaged. Nuclei/fibre were scored as nuclei within laminin rings at three trunk positions in three fish of each genotype. The data are presented as the mean of averaged values from each individual fish. For digital whole section reconstruction several images where taken on iRiS™ Digital Cell Imaging System, using ×4 objective, and merged using Photoshop CS5.1.

**NADH Tetrazolium Reductase**. NADH-TR protocol was adapted from (https://neuromuscular.wustl.edu/pathol/histol/nadh.htm). Briefly, 15 μm unfixed cryosections, from three 120 dpf adult male length-matched fish of each genotype, were incubated in a 1:1 solution of NBT (Nitro-blue tetrazolium, 2 mg/ml, N6876, Sigma) and NADH (1.6 mg/ml, N8129, Sigma) in 0.05 M Tris HCl pH 7.6 at RT for 2 h. Sections were then washed three times with deionized water (dH₂0), serially immersed in acetone:water 30%, 60%, 90%, 60%, 30%, ×3 dH₂0, glycerol mounted and imaged with Axiophot microscope (Zeiss) equipped with Olympus DP-70 camera.

**Isolation and culture of zebrafish MPCs from adult tissue**. Isolation and culture of zebrafish adult muscle fibres was adapted from ref.[91]. Briefly, adult fish were culled in high dose tricaine, washed in PBS, then 70% ethanol, eviscerated and skinned. Trunk muscle was incubated in 0.2% Collagenase (C0130, Sigma Aldrich), 1% Penicillin/Streptomycin DMEM at 28.5 °C for at least 2 h. Single muscle fibres were released by trituration using heat-polished glass pipettes and washed three times with DMEM. 90–100 myofibres per fish were plated on Matrigel (Invitrogen) coated 24-well plates and cultured in 20% Foetal Bovine Serum in 1% Penicillin/Streptomycin/ DMEM for 7 days. Cells were washed twice with PBS to remove muscle fibres and induced to differentiate in 2% Horse Serum 1% Penicillin/Streptomycin/ DMEM for 5 days at 28.5 °C in 5% CO₂ with medium change every 48 h, then fixed with 4% PFA, processed for immunofluorescence and imaged at 20X using an Axiovert 200 M microscope (Zeiss). At least five random fields were acquired in each of three technical replicates on each fish. Six 15 months old adult male (three *myog*^kg125/+ heterozygotes and three *myog*^kg125 mutants) were dissected for the analysis. Differentiation index = nuclei in MyHC⁺ myocytes (MF20 and A4.1025) ×100/nuclei in desmin⁺ cells. Fusion index = Nuclei in myocytes with ≥ 2 nuclei ×100/ total nuclei in MyHC⁺ myocytes.

**Statistical analyses**. Quantitative analysis on images was performed with Fiji software (NIH, Fiji.sc). Statistical analyses used GraphPad (Prism 6) for unpaired two-tailed Student's *t* test or Statplus:mac v5 for ANOVA with Bonferroni or Tukey post-hoc tests to assess significant differences between mutant and sibling groups, unless otherwise stated. χ² test was used to analyse difference between distributions using raw values. All data are expressed as mean ± standard error of the mean (SEM). Unless otherwise stated, numbers on columns represent number of fish analysed. *p* values for rejection of the null hypothesis of no difference between groups are indicated above columns.

**Data availability**
The authors declare that all the data supporting the findings of this study are available within the Article and its Supplementary Information files or from the corresponding author upon reasonable request.

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

## Acknowledgements

We are grateful to all members of the Hughes lab for advice and to Bruno Correia da Silva and his staff for care of the fish. We thank Nicolas Figeac for helpful discussion and advice on single fibre isolation protocol setup and Daniel S. Osborn for initial cloning of *myogenin* CDS into *hsp70-4:MyogCDS-IRES-NLSmGFP6* plasmid. This work is supported by grants from the Medical Research Council to S.M.H. (MRC Programme Grants G1001029 and MR/N021231/1) and P.S.Z. (MR/P023215/1), to Y.H. and S.M.H. from the British Heart Foundation (PG/14/12/30664) and to P.S.Z. from Muscular Dystrophy UK (RA3/3052), Association Française contre les Myopathies (AFM17865) and the FSH Society (FSHS-82013-06 and FSHS-82017-05).

## Author contributions

M.G. and S.M.H. conceived the project and designed the experiments. Y.H. designed CRISPR strategy and generated *myog* mutant lines. M.G. performed all the experiments and analysis. S.B. designed and performed ChIP-qPCR assay, contributed to embryo genotyping and sample preparation for qPCR. M.G., H.P.O.Q. and P.S.Z. established the protocol for single fibre isolation and culture of zebrafish adult MPCs. S.M.H. and M.G. wrote the paper with contributions from all other authors.

## Additional information

**Competing interests:** The authors declare no competing interests.

