## [Peer Review File · Nature Communications]

Reviewers' Comments:

Reviewer #1:

Remarks to the Author:

The manuscript by Ganassi et al details an investigation of the myogenic bHLH transcription factor, Myogenin, in muscle development and differentiation. Studies using mouse models have indicated that Myogenin plays an essential role in the differentiation of myoblasts into myofibres during embryonic development. Myogenin, in contrast to MyoD, Myf5 or MRF4, is necessary for viability as removal of Myogenin during embryogenesis leads to perinatal death due to defective muscle differentiation. Compatible with a role in muscle differentiation, microarray studies implicate Myogenin in regulating such genes as actinin and titin, depending on the muscle group and period of removal. Depletion of Myogenin after birth, however, doesn't block muscle differentiation as seen during embryogenesis but reduces organismal size and myofiber size. Moreover, loss of Myogenin has also been linked to alterations in muscle metabolism, regulating genes involved in oxidative phosphorylation and glycolysis. Linked to this regulation of metabolism is, for example, the observation that loss of Myogenin in mdx mice leads to an increase in exercise capacity. Nevertheless, how Myogenin achieves these various roles remains unclear.

Using a zebrafish model, the authors report a new, evolutionarily conserved function for Myogenin in regulating myocyte fusion and challenge the interpretations/conclusions of several studies. The authors generated two new null alleles of Myogenin using CRISPR/Cas9. For both new alleles, a stop codon was introduced at amino acid 37, which would lead to a truncated protein without the basic and HLH domains. Using these new alleles, the authors describe phenotypes associated with Myogenin loss. They find that Myogenin is dispensable for the initial phases of myogenesis: specification and formation of functional fast and slow muscle fibres occurred normally. Motor function was found to be similar between the mutant and siblings at 5dpf. Nevertheless, closer examination revealed that there was a reduction in myotome size at 1 and 2 dpf. This reduction in myotome size was due to a decrease in the size of the individual fast myofibers, despite an increase in numbers. The majority of fast fibres had only one nucleus/fibre, indicating a fusion block. Additional experiments revealed that Myogenin was responsible for regulating key proteins involved in myocyte fusion, including Myomaker, Myomixer-Minion-Myomerger, and Jam3b. Interestingly, the authors also uncovered some residual myocyte fusion that occurred in the absence of Myogenin; this fusion, located deep in the myotome near the neural tube, was dependent on Hedgehog signaling. Lastly, while Myogenin null fish are viable, the authors present data that indicate that the mutant fish are smaller than controls (a phenotype found when Myogenin is removed postnatally in mouse). Despite myofibre formation being detected in the adult, these myofibres had fewer nuclei than the controls, consistent with a continued block/reduction in fusion. These smaller fibres were correlated with the overall smaller size of the organism. Thus, a common role for Myogenin in the embryo and in the adult centers on its control of myocyte fusion, which the authors argue is the evolutionarily conserved role for Myogenin.

No doubt, this work provides new insights to the role of Myogenin in muscle development, that of regulating sets of genes involved in myocyte fusion. This work challenges the prevailing view of Myogenin's function during muscle formation, that of regulating "all" aspects of muscle differentiation including fusion, myofibril formation, innervation, etc during embryogenesis. Moreover, the authors suggest that this primary role for Myogenin continues throughout the life of the organism and discusses how work in *Xenopus*, chick and mouse could be interpreted in this light. This is exciting work and raises interesting questions about myogenin and differentiation. Nevertheless, there are a several concerns with the data and conclusions as presented.

Fig1. New Crispr alleles

- a. While the data are consistent with morpholino kd, it would be advised that the authors provide evidence that there are no off target effects associated with the new Myogenin alleles generated with CRISPR/Cas9. It appears that the alleles were made with the same guide RNAs and at the same time. Rescue with Myogenin would be advised.
- b. Are their changes in MyoD levels (or MRF4 or Myf5 levels) in the null alleles? Is there any

compensation ?

- c. Controls – the “sibs” in each experiment: are they myogenin +/- or +/+? Do the hets have any phenotypes?
- d. Could the authors describe the phenotype of the hypomorphic allele myog-fh265?

Fig2. Embryonic myoblast differentiation and muscle function

- a. Line 105.. revise to ,“Thus without myogenin, specification and early development of slow and fast muscle appears normal.
- b. The authors show no difference in slow muscle differentiation (ie numbers and placement) at the timepoints analyzed. What about later stages? Also, given that Myogenin in mouse is reported to have a role in muscle metabolism, particularly in regulating oxidative phosphorylation of slow muscles, have the authors checked for that role either in the larva or in adult slow muscle? Would be very interesting if that role was also conserved. Moreover, there are publications that suggest a switch from oxidative phosphorylation to glycolysis during differentiation ,and particularly fusion, which would be interesting to link to Myogenin.
- b. Panel 2E (specifically: line 108-109, “ ...fast muscle appearedand slightly reduced in extent”). Quantify “slightly reduced in extent”...
- c. Muscle function: Studies with loss of function of myogenin in mouse (adult) leads to enhanced exercise capacity and increased muscle fatigue resistance. It seems that to see this phenotype in this model, the swimming tests given would have to be more challenging (swimming in a current perhaps). Have the authors tried such experiments in the larvae or adult?

Fig 3 Muscle size

- a. Related question relevant to the observation that more fibres of smaller size are formed due to loss of myocyte fusion: Do more fibres packed in the myotome limit the growth of those fibres? If one were to ablate a number of them, would the remaining myogenin-/- fibres increase their diameters? In other words, do the fibres have to be bi- or tri- etc nucleated to support the appropriate growth of the fibers ?
- b. When do the slow muscle fibres fuse in the adult? Or do they? How do their sizes compare to the fast fibres?

Fig 4. Myogenin is required for normal myocyte fusion

- a. Panels 4 E,F -- Do these pie charts include the slow and the fast muscle fibres? I assume that is the 33% shown is the slow?
- b. What occurs after 6dpf? (Line 147-148 “This phenotype persisted at least until 6dpf”)
- c. Ideally the authors would show that overexpression would lead to increased fusion as was published with Myomaker (Zhang and Roy, 2017), and myomixer/minion/myomerges (Shi et al., 2017). This construct could also be used for rescue experiments.

Figure 5: Expression of essential cell fusion genes

- a. Please quantify reductions in Fig. S4A,B
- b. It would be helpful to have the percent reduction stated in the text for each of the genes where there is reduction.
- c. Given published data on Myomaker and myomixer/minion/myomerges in zebrafish, does the extent of reduction fit with the phenotypes detected in the myogenin mutants?
- d. While the reduction in RNA levels and Chip data are convincing, it would be convincing that those sites are absolutely necessary if results using a reporter with mutations in the E boxes were included.

Figure 6 Hedgehog signaling drives residual fusion and Myomaker expression.

- a. Please quantify Myomaker reduction.
- b. Why is there residual Myomaker expression in the drug treated animals?

Figure 7 Adult myogenesis

- a. Similar to earlier development, there are more, but smaller, fibres in the adult. However the

overall size of the adult is smaller. It is not clear in the text how the authors connect these phenotypes. One could suggest that there are myokines released from the muscle that coordinate organismal growth. OR ?

b. Panel F-H, on which day are these analyses done? Is there a reason not to quantify more time points to show the progression (or lack of change) in the myogenin mutants compared to control? These data would provide support for a role for myogenin in "homeostasis". Also, the first column in panel H – 0 nuclei per fibre? Was this just a particular section? Could the number of nuclei per entire fibre be provided? What are typical numbers of myonuclei in fast and slow muscle at this point in adult life?

c. line 199-201. This sentence is unclear. "The total number of fibres in single cross-sections of a mutant (4097) and a sib (3657) were similar, and were approximately 40-fold those in the larvae, indicating that fiber formation had persisted." I understand that the number of fibres in each background increased (40 fold), but since the total numbers were the same in mutant and sib, doesn't that mean that the increase seen in the myogenin mutant larvae (about twice the amount) now shows decreased denovo formation?

Discussion

a. Terminal differentiation usually includes "becoming multinucleate" for a lot of muscle people. Please define that your particular definition and what it includes such as becoming multinucleate, elongating, attaching to segment border, innervation, myofibrils assembly... The use of this phrase is confusing in the discussion.

b. Desmin discussion, please reference

c. "Later steps in myocyte differentiation" (line 232) = fusion.

d. Line 274 The data shown indicates that in the absence of Myogenin, all "fast" myocytes differentiate into a muscle cells whereas in the controls that express myogenin, some cells are selected that are able to fuse with other cells. The authors argue that this rules out the idea of founder cells in fish. Does it? Perhaps in the absence of myogenin, all cells are "founder cells".

Methods:

In addition to comments above, please clearly indicate in the methods which axis level was used for confocal cross sections.

General issue: The writing is confusing in many places, making it difficult to read; the manuscript would benefit from close reading and editing.

Reviewer #2:

Remarks to the Author:

In this very interesting and important paper by Ganassi et al the function of zebrafish myogenin is genetically dissected. Myogenin is a member of the bHLH class of myogenic regulators that have been previously extensively dissected in amniote myogenesis, although Myogenin remains the least understood member of this class. Previous analyses by these authors had examined an ENU induced hypomorph allele, which revealed little about its embryonic and larval function. In order to further examine the function a CRISPR/Cas9 null allele was created. In contradiction to the function commonly ascribed to mammalian Myog, the authors describe that Myog is dispensable for myoblast terminal differentiation and instead is required for the fusion of differentiating myocytes. Lack of Myog prevents expression of proteins involved in cell fusion including the critical regulator Myomaker. A small amount of fusion does appear to occur deep myotome depending upon Hedgehog signalling, suggesting the existence of at least two independent pathways regulating vertebrate myocyte fusion.

General Comments.

This is beautifully documented, presented and well written paper that makes a very strong contribution to our understanding of this basic developmental process. Myocyte fusion is a very

topical and central theme in current myogenesis studies, with the recent discoveries of cell biological regulators such as Myomaker that facilitate this process. But how muscle cells initiate the fusion process and how fusion genes expression is regulated is largely unknown. An important question is how the fusion process related to the core regulators of myogenesis. This report provides an important piece of the puzzle of how this may occur. It also prompts a reevaluation of our understanding of Myogenin function more generally.

Specific Comments.

1. While important to document that muscle differentiation is normal, and the authors do an excellent and thorough job of this, Figure 2 is a lot of negative data that perhaps could mostly be placed in supplementary data?
2. Could the authors clarify the role of myogenin in slow muscle? It is mentioned that it is expressed at a lower level than in fast fibres. Is this the reason that slow fibres don't fuse? Could you induce slow muscles to fuse if you increase myogenin expression in slow cells. If not this would suggest the existence of another pathway actively preventing fusion to occur in slow cells.
3. The authors suggest for one of the central 4 findings is "that Myog is required for normal myogenesis throughout life and that its loss leads to poor muscle and whole body growth, suggesting a persistent functional deficit." This is an important point but I am not sure that the data presented supports this conclusion. Could not the deficits that are presented result from a developmental requirement for myogenin in the embryo and larvae and the defects presented in adults result from developmental deficits? In order to make this important point the authors would need show that fusion deficits also occur in fibres that form in juveniles and adults. Perhaps a BRDU pulse chase in juveniles and adults to show a lack of fusion in BRDU containing adult muscle fibres? Methods for the isolation of adult muscle fibres and nuclei counts have been previously published. I contend this technically simple experiment would add a lot to the interpretation of the results, and provide evidence that myogenin is required for fusion through out life. It would also support the central role of myogenin in fusion generally.
4. In the discussion the authors rightly examine the broader murine Myog literature looking for suggestions that the interpretation of this knockout as showing a block in myogenesis may be incorrect. Could the authors state directly if fusion has ever been examined in the mutant or in culture muscle cells derived from the mutant. I think this would be important to say.

Reviewer #3:

Remarks to the Author:

This manuscript reports the phenotype of a myogenin loss of function allele in zebrafish. Myogenin has traditionally been depicted as a necessary transcription factor for terminal differentiation, although there were various inconsistencies for its precise action in both mouse and zebrafish myogenesis. Somewhat surprisingly, the authors show that myog is not required for terminal differentiation per se, as the fish that lack myog exhibit differentiated myocytes and functional muscle. Instead their data indicate myog is more specifically required for the fusion program. They also provide some evidence for a myog-independent fusion pathway, through Hh. Each of these concepts are novel and would be informative for the field. The manuscript is well-written and the quality of the data are excellent. I have the following suggestions to improve the manuscript:

1. One of the interpretations of the paper is that that myog is required throughout life, however it is not clear if the data support this assertion. The main problem is that any phenotype in the adult kg125 fish could be due to loss of myog during the embryonic stage, and not loss of myog specifically in the adult.
2. What are the levels of the myog protein? It is surprising that myog mRNA is only reduced 30-40%, and I realize that the mutation is a premature stop codon, however the only assessment of myog protein levels is by immunostaining which is not quantified. Full assessment of myog protein

in all of the various lines would help solidify interpretation.

3. The authors perform experiments on either kg125 or kg128 and there is no obvious rationale for choosing one over the other. Maybe the lines are so similar that they can be used interchangeably, however it would be nice if the authors showed this definitively early in the paper. Fig. 1 now shows in situ for kg128, but qPCR and immuno for kg125. Direct side-by-side comparison of myog expression and general phenotype would be helpful.

4. Maybe this is beyond the scope of the current paper, but it would be interesting to test if overexpression of myomaker would rescue the fusion phenotype in myog null fish.

5. How the authors define replicates is not clear. For instance, figure 1C indicates 'four independent experiments'. Does this mean four independent fish were analyzed or one fish was analyzed four independent times.

Response to Referees

Reviewer #1 (Remarks to the Author):

The manuscript by Ganassi et al details an investigation of the myogenic bHLH transcription factor, Myogenin, in muscle development and differentiation. Studies using mouse models have indicated that Myogenin plays an essential role in the differentiation of myoblasts into myofibres during embryonic development. Myogenin, in contrast to MyoD, Myf5 or MRF4, is necessary for viability as removal of Myogenin during embryogenesis leads to perinatal death due to defective muscle differentiation. Compatible with a role in muscle differentiation, microarray studies implicate Myogenin in regulating such genes as actinin and titin, depending on the muscle group and period of removal. Depletion of Myogenin after birth, however, doesn't block muscle differentiation as seen during embryogenesis but reduces organismal size and myofiber size. Moreover, loss of Myogenin has also been linked to alterations in muscle metabolism, regulating genes involved in oxidative phosphorylation and glycolysis. Linked to this regulation of metabolism is, for example, the observation that loss of Myogenin in mdx mice leads to an increase in exercise capacity. Nevertheless, how Myogenin achieves these various roles remains unclear.

Using a zebrafish model, the authors report a new, evolutionarily conserved function for Myogenin in regulating myocyte fusion and challenge the interpretations/conclusions of several studies. The authors generated two new null alleles of Myogenin using CRISPR/Cas9. For both new alleles, a stop codon was introduced at amino acid 37, which would lead to a truncated protein without the basic and HLH domains. Using these new alleles, the authors describe phenotypes associated with Myogenin loss. They find that Myogenin is dispensable for the initial phases of myogenesis: specification and formation of functional fast and slow muscle fibres occurred normally. Motor function was found to be similar between the mutant and siblings at 5dpf. Nevertheless, closer examination revealed that there was a reduction in myotome size at 1 and 2 dpf. This reduction in myotome size was due to a decrease in the size of the individual fast myofibers, despite an increase in numbers. The majority of fast fibres had only one nucleus/fibre, indicating a fusion block. Additional experiments revealed that Myogenin was responsible for regulating key proteins involved in myocyte fusion, including Myomaker, Myomixer-Minion-Myomerger, and Jam3b. Interestingly, the authors also uncovered some residual myocyte fusion that occurred in the absence of Myogenin; this fusion, located deep in the myotome near the neural tube, was dependent on Hedgehog signaling. Lastly, while Myogenin null fish are viable, the authors present data that indicate that the mutant fish are smaller than controls (a phenotype found when Myogenin is removed postnatally in mouse). Despite myofibre formation being detected in the adult, these myofibres had fewer nuclei than the controls, consistent with a continued block/reduction in fusion. These smaller fibres were correlated with the overall smaller size of the organism. Thus, a common role for Myogenin in the embryo and in the adult centers on its control of myocyte fusion, which the authors argue is the evolutionarily conserved role for Myogenin.

No doubt, this work provides new insights to the role of Myogenin in muscle development, that of regulating sets of genes involved in myocyte fusion. This work challenges the prevailing view of Myogenin's function during muscle formation that of regulating "all" aspects of muscle differentiation including fusion, myofibril formation, innervation, etc during embryogenesis. Moreover, the authors suggest that this primary role for Myogenin continues throughout the life of the organism and discusses how work in *Xenopus*, chick and mouse could be interpreted in this light. This is exciting work and raises interesting questions about myogenin and differentiation. Nevertheless, there are a several concerns with the data and conclusions as presented.

We are delighted that the Reviewer finds our work exciting and interesting and that it provides new insights into the role of Myogenin in muscle development. The Reviewer asks a very large number of interesting questions. We have responded to all those that bear directly on the clarity or interpretation of our reported findings with new experimental data and/or discussion. However, a number of the points raised, while interesting, do not bear on the interpretation or significance of our conclusions. In a number of these cases, we have declined to do further experiments because it would unreasonably delay the return of the manuscript.

Fig1. New Crispr alleles

a. While the data are consistent with morpholino kd, it would be advised that the authors provide evidence that there are no off target effects associated with the new Myogenin alleles generated with CRISPR/Cas9. It appears that the alleles were made with the same guide RNAs and at the same time. Rescue with Myogenin would be advised.

In Fig. 4i,j and Fig S4 we now provide evidence of rescue of fusion by Myogenin over-expression in the mutant by DNA injection of a *myog:MyogCDS-IRES-GFP* vector. Interestingly, the results suggest Myog is only required in one partner cell.

b. Are their changes in MyoD levels (or MRF4 or Myf5 levels) in the null alleles? Is there any compensation?

In Fig. 2A, we now show by qRT-PCR that *mrf4* and *myf5* are significantly downregulated in *myogenin* mutants (54% and 40%, respectively). In contrast, *myod* RNA level is unaltered. This reduces the possibility of a compensatory effect by other MRFs minimizing the defects observed.

c. Controls – the “sibs” in each experiment: are they myogenin +/- or +/-? Do the hets have any phenotypes?

The hets have no significant phenotype, which is why we pooled them with wt into ‘sibs’. This is now stated more clearly on pg 3, lines 107-109 and pg 11 lines 447-450. We also show that Myogenin protein level is not significantly different between het and wild-type siblings (Fig. S1a).

d. Could the authors describe the phenotype of the hypomorphic allele *myog-fh265*?

The lack of phenotype has been partially described in Hinits et al 2011 and more is presented in Figs 7b, S2c-e, S5a and S6a. We have now expanded our description of the lack of phenotype in the *myog^{fh265}* allele on pg 4, ln 144-146.

Fig2. Embryonic myoblast differentiation and muscle function

a. Line 105.. revise to ,”Thus without myogenin, specification and early development of slow and fast muscle appears normal.

Done. Now on pg 3, ln 116.

b. The authors show no difference in slow muscle differentiation (ie numbers and placement) at the timepoints analyzed. What about later stages?

We have now looked at adult and find reduced area of the slow muscle region in sections from adult mutants, which is now highlighted in Fig. 7i and on pg 6, ln 239-240.

Also, given that Myogenin in mouse is reported to have a role in muscle metabolism, particularly in regulating oxidative phosphorylation of slow muscles, have the authors checked for that role either in the larva or in adult slow muscle? Would be very interesting if that role was also conserved.

We agree with the Reviewer. We have now performed NADH tetrazolium reductase staining on mutants and sibs and this is reported in Fig. 7i and pg. 6,ln 239-242. We observe an overall reduction in NADH-TR stain in the slow/intermediate fibre region.

Moreover, there are publications that suggest a switch from oxidative phosphorylation to glycolysis during differentiation, and particularly fusion, which would be interesting to link to Myogenin.

We agree that these are interesting questions, but we do not think they bear on the validity or importance of the findings that we describe. Moreover, this as a controversial subject; some papers claim muscle cells become more oxidative during terminal differentiation, not

less (e.g. Sin et al 2016 *Autophagy* 12:369; Leary et al 98 *BBA* 1365:522). Therefore, we have not examined changes in metabolism during myoblast differentiation in the current manuscript.

b. Panel 2E (specifically: line 108-109, "...fast muscle appearedand slightly reduced in extent"). Quantify "slightly reduced in extent"...

The reduction in extent is quantified in Fig. 3b,c,f-i.

c. Muscle function: Studies with loss of function of myogenin in mouse (adult) leads to enhanced exercise capacity and increased muscle fatigue resistance. It seems that to see this phenotype in this model, the swimming tests given would have to be more challenging (swimming in a current perhaps). Have the authors tried such experiments in the larvae or adult?

We have not done such experiments. However, we do not think they bear on the validity or importance of the findings that we describe. What we have now done to address the issue raised is analyse swimming behaviour in a viscous methyl cellulose (MC) solution and find that, although MC decreases swimming performance in both sibs and mutants, behaviour of mutant larvae is not significantly different from that of siblings. These studies are now shown in Fig. 2i and described on pg 3, ln 124-126. In addition, we have grown larvae in methyl cellulose, which is known to stress defective muscle. We find that both siblings and mutant larvae retain good muscle morphology (Fig. S1f). This result is now reported on pg 4, ln 126-128.

Fig 3 Muscle size

a. Related question relevant to the observation that more fibres of smaller size are formed due to loss of myocyte fusion: Do more fibres packed in the myotome limit the growth of those fibres? If one were to ablate a number of them, would the remaining myogenin^{-/-} fibres increase their diameters? In other words, do the fibres have to be bi- or tri- etc nucleated to support the appropriate growth of the fibers ?

These are interesting questions and clearly expressed. The manuscript contains a partial answer: the residual medial multinucleate fibres in mutants are indeed bigger than adjacent mononucleate fibres and are therefore not limited by packing (Figs 3e, 6c). In Roy et al (2017) we also showed that in *myod* mutants, which have fewer fibres, the remaining fibres grow larger. Although we did not specifically look at mononucleate fibres in those experiments, we did show that fusion of extra nuclei to the remaining fibres did not account for the larger growth. Moreover, as we have shown (Pipalia et al 2016), larval muscle rapidly regenerates from stem cells, so this regeneration would also need to be inhibited to perform the requested assay. Whether or not the remaining fibres grew larger, we do not think the result could significantly alter the conclusions or significance of the current manuscript.

b. When do the slow muscle fibres fuse in the adult? Or do they? How do their sizes compare to the fast fibres?

As we previously published (Roy et al., 2017 Fig. 1A) and show in new Fig. 7i, slow fibres are smaller than fast in the adult. If and when fusion occurs is not known and is interesting. But, again, we do not think that knowing the answers could alter the conclusions or significance of the current manuscript.

Fig 4. Myogenin is required for normal myocyte fusion

a. Panels 4 E,F -- Do these pie charts include the slow and the fast muscle fibres? I assume that is the 33% shown is the slow?

No, the charts show only data for fast fibres: some fast fibres are mononucleate at the stage analysed. This is now stated in the legend. Slow fibres are additionally present in wild type numbers in mutant. This is now explicitly stated in the legend to Fig. 4 on pg 17 In 656.

b. What occurs after 6dpf? (Line XX147-148 “This phenotype persisted at least until 6dpf”)

The answer to this question is revealed in the expanded Figs 7, 8 and S6.

c. Ideally the authors would show that overexpression would lead to increased fusion as was published with Myomaker (Zhang and Roy, 2017), and myomixer/minion/myomergner (Shi et al., 2017). This construct could also be used for rescue experiments.

We have now performed this experiment and added it as Figs 4i,j and S4. It is described on pg 5, In 180-181. No more fusion was observed when Myog was overexpressed in wild type muscle.

Figure 5: Expression of essential cell fusion genes

a. Please quantify reductions in Fig. S4A,B

We do not think quantifying the changes/lack of change by qRT-PCR is necessary. A minor change or lack of change in *myog*^{fh265} would not change our conclusions. The reduction in *myog* and *mymk* mRNAs in *myod* mutants would clearly simply correlate with the reduced myotome volume that we already published (Hinits et al 2011; Roy et al 2017). However, we have quantified the reduction of *mymk* expression in Cyclopamine-treated *myog*^{kg128} and *myog*^{kg125} and their wild-type sibs. The analysis in Fig. S5d shows that CyA significantly reduces *mymk* expression also in wt (22%) and is described in pg 6, In 217-220.

b. It would be helpful to have the percent reduction stated in the text for each of the genes where there is reduction.

These are now included.

c. Given published data on Myomaker and myomixer/minion/myomergner in zebrafish, does the extent of reduction fit with the phenotypes detected in the myogenin mutants?

This is an interesting question which we now discuss on pg 8, In 303-313. Bottom line: it is not clear because the studies on Myomaker and myomixer/minion/myomergner function in zebrafish did not show any quantitative data.

d. While the reduction in RNA levels and Chip data are convincing, it would be convincing that those sites are absolutely necessary if results using a reporter with mutations in the E boxes were included.

Zhang and Roy (2017) already showed that a 3.3 kb *mymk* promoter element contains 23 E-boxes. Our ChIP data focus attention on two in particular, most importantly the most promoter proximal E-box. Millay et al (2014) showed that a 5' E-box at -41 bp is essential for expression in muscle from a 1.7 kb murine *Mymk* promoter fragment. Luo et al (2015) showed that the more proximal of two E-boxes in the proximal 0.6 kb of chicken *Mymk* is essential in myotube luciferase assays. While we agree that the suggested experiment would strengthen our conclusion, it would not prove that Myogenin acts there because of the co-expression of other MRFs. Moreover, showing that an E-box is required for higher reporter gene expression in a construct out of the endogenous locus does not provide proof of function in the endogenous locus. We therefore have not performed this experiment, preferring in future work to use genome editing to mutate the endogenous locus. Such studies require several generations of fish breeding and go beyond the current manuscript.

Figure 6 Hedgehog signaling drives residual fusion and Myomaker expression.
a. Please quantify Myomaker reduction.

We have now performed qRT-PCR to quantify the *mymk* mRNA reduction in sorted CyA-treated sibs and mutants. This is shown in Fig. 6b. The answer is a 54% reduction. We have also expanded the quantification of *mymk* reduction on *myog*^{kg125} and *myog*^{kg128} siblings. This result is now reported in Fig. S5c,e and described on pg 6, ln 217-221.

b. Why is there residual Myomaker expression in the drug treated animals?

We cannot explain this, but it is the case. Perhaps the cyclopamine was not fully effective. We now discuss this on pg 9, ln 363-364.

Figure 7 Adult myogenesis

a. Similar to earlier development, there are more, but smaller, fibres in the adult. However the overall size of the adult is smaller. It is not clear in the text how the authors connect these phenotypes. One could suggest that there are myokines released from the muscle that coordinate organismal growth. OR ?

We agree that myokines are one possibility, but a simpler one in our view is that fish with a minor defect in muscle function fail to feed like siblings and therefore grow more slowly. Nevertheless, on top of any such effect, our evidence suggests that muscle is smaller even when overall body size is taken into account. This is one reason why we favour a specific and persistent defect in myogenesis. We have now clarified the existence of such alternatives in the Discussion pg 10 ln 381-386.

b. Panel F-H, on which day are these analyses done?

120 dpf, now clarified in Methods pg 14 ln 544 and in Fig legends.

Is there a reason not to quantify more time points to show the progression (or lack of change) in the myogenin mutants compared to control? These data would provide support for a role for myogenin in "homeostasis".

This can and is being done, but requires ageing of fish and analysis of numerous independent cohorts/lays to ensure reproducibility of the data. We prefer not to delay publication until this is completed, but instead to include it in a planned future publication on the adult phenotype, as described further below. We now provide preliminary evidence that the muscle size defect is larger at 15 months of age in Fig. S6c.

Also, the first column in panel H – 0 nuclei per fibre? Was this just a particular section?

This is indeed nuclei/fibre cross section (fibre profile). We have now clarified this in the legend pg 18, ln 703-705.

Could the number of nuclei per entire fibre be provided? What are typical numbers of myonuclei in fast and slow muscle at this point in adult life?

These are important questions. Detailed analysis of the adult phenotype and its progression is on-going in our laboratory and requires development of reproducible methods of single fibre analysis and unbiased quantification in the zebrafish.

c. line 199-201. This sentence is unclear. "The total number of fibres in single cross-sections of a mutant (4097) and a sib (3657) were similar, and were approximately 40-fold those in the larvae, indicating that fiber formation had persisted." I understand that the number of fibres in each background increased (40 fold), but since the total numbers were the same in mutant and sib, doesn't that mean that the increase seen in the myogenin mutant larvae (about twice the amount) now shows decreased denovo formation?

No, it does not. The Reviewer has read and understood our sentence correctly. However, as these fibres are counted in cross sections the number of fibres counted is not necessarily a measure of the total number in each myotome.

Discussion

a. Terminal differentiation usually includes "becoming multinucleate" for a lot of muscle people. Please define that your particular definition and what it includes such as becoming multinucleate, elongating, attaching to segment border, innervation, myofibrils assembly... The use of this phrase is confusing in the discussion.

In our view, the definition of terminal differentiation in developmental and cell biology is 'permanent exit from the cell cycle and correlated activation of tissue-specific genes', as stated in Alberts et al, 5th Edition pg 1103. However, an informal survey of colleagues has revealed that the Reviewer is not alone; there is much confusion. We have therefore included a specific clarification of our definition of this term in the Introduction pg 1 In 40-41.

b. Desmin discussion, please reference

Done.

c. "Later steps in myocyte differentiation" (line 232) = fusion.

Modified as requested.

d. Line 274 The data shown indicates that in the absence of Myogenin, all "fast" myocytes differentiate into a muscle cells whereas in the controls that express myogenin, some cells are selected that are able to fuse with other cells. The authors argue that this rules out the idea of founder cells in fish. Does it? Perhaps in the absence of myogenin, all cells are "founder cells".

This is a really good point that had occurred to us but somehow got cut out during manuscript editing. We now state this option explicitly on pg 9, In 349-351. Many thanks for noticing the omission.

Methods:

In addition to comments above, please clearly indicate in the methods which axis level was used for confocal cross sections.

Done. It is also mentioned in the legends.

General issue: The writing is confusing in many places, making it difficult to read; the manuscript would benefit from close reading and editing.

We have gone through the manuscript and hope the Reviewer now finds it improved.

Reviewer #2 (Remarks to the Author):

In this very interesting and important paper by Ganassi et al the function of zebrafish myogenin is genetically dissected. Myogenin is a member of the bHLH class of myogenic regulators that have been previously extensively dissected in amniote myogenesis, although Myogenin remains the least understood member of this class. Previous analyses by these authors had examined an ENU induced hypomorph allele, which revealed little about its embryonic and larval function. In order to further examine the function a CRISPR/Cas9 null allele was created. In contradiction to the function commonly ascribed to mammalian Myog, the authors describe that Myog is dispensable for myoblast terminal differentiation and instead is required for the fusion of differentiating myocytes. Lack of Myog prevents expression of proteins involved in cell fusion including the critical regulator Myomaker. A small amount of fusion does appear to occur deep myotome depending upon Hedgehog signalling, suggesting the existence of at least two independent pathways regulating vertebrate myocyte fusion.

General Comments.

This is beautifully documented, presented and well written paper that makes a very strong contribution to our understanding of this basic developmental process. Myocyte fusion is a very topical and central theme in current myogenesis studies, with the recent discoveries of cell biological regulators such as Myomaker that facilitate this process. But how muscle cells initiate the fusion process and how fusion genes expression is regulated is largely unknown. An important question is how the fusion process related to the core regulators of myogenesis. This report provides an important piece of the puzzle of how this may occur. It also prompts a reevaluation of our understanding of Myogenin function more generally.

We are gratified by the enthusiasm for our work shown by the Reviewer.

Specific Comments.

1. While important to document that muscle differentiation is normal, and the authors do an excellent and thorough job of this, Figure 2 is a lot of negative data that perhaps could mostly be placed in supplementary data?

As the Reviewer acknowledges, the lack of a terminal differentiation defect is a major new aspect of our work that we feel it is important to document this thoroughly in the main text. Sadly, readers frequently consider Supplementary Data in papers to be of secondary importance and possibly lower quality, although this is not the case here. We believe our careful single cell resolution analyses are a major strength of the paper and prefer to retain this data in the main text.

2. Could the authors clarify the role of myogenin in slow muscle? It is mentioned that it is expressed at a lower level than in fast fibres. Is this the reason that slow fibres don't fuse? Could you induce slow muscles to fuse if you increase myogenin expression in slow cells. If not this would suggest the existence of another pathway actively preventing fusion to occur in slow cells.

We have now addressed this issue experimentally by over-expressing Myog mosaically in mutants and siblings as described on pg. 5, in 171-186 and in Figs 4i,j and S4a-f. Over-expression in sibling slow fibres did not induce fusion. However, over-expression in mutant fast fibres does rescue fusion, showing the expressed Myogenin is active in fast fibres.

3. The authors suggest for one of the central 4 findings is "that Myog is required for normal myogenesis throughout life and that its loss leads to poor muscle and whole body growth, suggesting a persistent functional deficit." This is an important point but I am not sure that the data presented supports this conclusion. Could not the deficits that are presented result from a

developmental requirement for myogenin in the embryo and larvae and the defects presented in adults result from developmental deficits? In order to make this important point the authors would need show that fusion deficits also occur in fibres that form in juveniles and adults. Perhaps a BRDU pulse chase in juveniles and adults to show a lack of fusion in BRDU containing adult muscle fibres? Methods for the isolation of adult muscle fibres and nuclei counts have been previously published. I contend this technically simple experiment would add a lot to the interpretation of the results, and provide evidence that myogenin is required for fusion through out life. It would also support the central role of myogenin in fusion generally.

The Reviewer is correct about the formal possibility that the persistent failure of muscle growth could arise from an early developmental defect. We have now addressed this issue directly by analysing fusion of cultured muscle cells from adult animals on pg 6, In 246-255 and in Fig. 8a-d and Fig S6f,g. To do this, we had to develop an entirely novel method for culturing zebrafish adult satellite cells, which constitutes a significant advance. The data show that a myocyte fusion defect persists. However, it must be acknowledged that, without an inducible Cre/lox-type experiment, this defect cannot formally be attributed to the cell autonomous role of Myogenin in the adult myocytes. This is now plainly stated in our Discussion (pg 10, In 393-394), although we think that a persistent embryonic defect that no longer depends on Myog expression in the myoblasts/myocytes is an unlikely alternative explanation.

4. In the discussion the authors rightly examine the broader murine Myog literature looking for suggestions that the interpretation of this knockout as showing a block in myogenesis may be incorrect. Could the authors state directly if fusion has ever been examined in the mutant or in culture muscle cells derived from the mutant. I think this would be important to say.

We now describe in more detail what has and has not been published on the murine *Myog* mutants on pg 7, In 267-278. The answer seems to be fusion has not been observed in vivo but does occur in vitro without Myog.

--

Reviewer #3 (Remarks to the Author):

This manuscript reports the phenotype of a myogenin loss of function allele in zebrafish. Myogenin has traditionally been depicted as a necessary transcription factor for terminal differentiation, although there were various inconsistencies for its precise action in both mouse and zebrafish myogenesis. Somewhat surprisingly, the authors show that myog is not required for terminal differentiation per se, as the fish that lack myog exhibit differentiated myocytes and functional muscle. Instead their data indicate myog is more specifically required for the fusion program. They also provide some evidence for a myog-independent fusion pathway, through Hh. Each of these concepts are novel and would be informative for the field. The manuscript is well-written and the quality of the data are excellent. I have the following suggestions to improve the manuscript:

We are pleased the Reviewer has a positive view of our work.

1. One of the interpretations of the paper is that that myog is required throughout life, however it is not clear if the data support this assertion. The main problem is that any phenotype in the adult kg125 fish could be due to loss of myog during the embryonic stage, and not loss of myog specifically in the adult.

The Reviewer is correct about the formal possibility that the persistent failure of muscle growth could arise from an early developmental defect. That is why we wrote 'Myog is required for normal myogenesis throughout life and that its loss leads to poor muscle and whole body growth, suggesting a persistent functional deficit'. Please note that we only 'suggested a persistent functional deficit'. We have now addressed this issue more directly by analysing fusion of cultured muscle cells from adult animals on pg 6, ln 246-255 and in Fig. 8a-d and Fig S6f,g. To do this, we had to develop an entirely novel method for culturing zebrafish adult satellite cells, which constitutes a significant advance. The data show that a myocyte fusion defect persists. However, it must be acknowledged that, without an inducible Cre/flox-type experiment, this defect cannot formally be attributed to the cell autonomous role of Myogenin in the adult muscle cells. This is now plainly stated in our Discussion (pg 10, ln 393-394), although we think that a persistent embryonic defect that no longer depends on Myog expression in the myoblasts/myocytes is an unlikely alternative explanation.

2. What are the levels of the myog protein? It is surprising that myog mRNA is only reduced 30-40%, and I realize that the mutation is a premature stop codon, however the only assessment of myog protein levels is by immunostaining which is not quantified. Full assessment of myog protein in all of the various lines would help solidify interpretation.

We have now quantified immunostaining on both lines, which is reported in Fig. 1d and Fig S1a. Western blots and more extensive analyses have not been possible because Santa Cruz Inc have stopped selling the antiserum we use to detect zebrafish Myog and to date no alternative has been found to work on zebrafish tissue.

3. The authors perform experiments on either kg125 or kg128 and there is no obvious rationale for choosing one over the other. Maybe the lines are so similar that they can be used interchangeably, however it would be nice if the authors showed this definitively early in the paper. Fig. 1 now shows in situ for kg128, but qPCR and immuno for kg125. Direct side-by-side comparison of myog expression and general phenotype would be helpful.

The Reviewer is correct that we have found no molecular or phenotypic differences between the two lines. We feel the dual analysis is a strength, rather than a weakness, of our work. Combining the Figures and Supplementary Figures, all major analyses are shown for both kg125 and kg128, with the exception of the new satellite cell culture analyses. Moreover, Reviewer 2 has asked us to reduce the data in the early part of the manuscript, rather than increase it. We have now performed ISH on a myog^{kg125} incross lay and quantitative fluorescence analysis on sorted mutants and sibs from both lines at 20ss and reveal similar reduction of mRNA and loss of Myog protein. As Fig 1c required

genotyping and pooling of individual embryos from four replicate lays, we have not repeated it. We hope we now strike an acceptable balance.

4. Maybe this is beyond the scope of the current paper, but it would be interesting to test if overexpression of myomaker would rescue the fusion phenotype in myog null fish.

We have not performed this experiment but have rescued with mosaic myog expression in Figs 4I,j and S4. We hope this is sufficient.

5. How the authors define replicates is not clear. For instance, figure 1C indicates 'four independent experiments'. Does this mean four independent fish were analyzed or one fish was analyzed four independent times.

In our view, analysing one fish four times is definitely NOT an 'independent experiment'. We took four separate lays of *myog*^{+/-} incross fish (probably, but not certainly, from different pairs of parents as animals were returned to a single tank after each use), separated DNA from RNA, genotyped each embryo, pooled mutant RNAs and, separately, wild types, and then made cDNA and performed qRT-PCR on the paired samples on four separate days. We have now clarified this in Fig. 1c legend pg 15, ln 603-606.

Reviewers' Comments:

Reviewer #1:

Remarks to the Author:

This is a re-review of the manuscript from Ganassi et al. The authors were extremely attentive to the critiques raised in my first review. I appreciate the efforts that went into the quantitation and to the improvement to the both the figures and the writing. The manuscript was a pleasure to read and will have significant impact on the muscle field.

The authors may want to consider shortening the title as follows:

Myogenin promotes myocyte fusion to balance muscle fiber number and size.

Reviewer #2:

Remarks to the Author:

The authors have addressed all my concerns with considerable new data. This important manuscript is ready for publication from my perspective.

Reviewer #3:

Remarks to the Author:

The authors have been very responsive to the concerns of all reviewers, therefore I have no other requests.